# Unified 2D-3D Discrete Priors for Noise-Robust and Calibration-Free Multiview 3D Human Pose Estimation

**Geng Chen**[*]   **Pengfei Ren**[*]   **Xufeng Jian**   **Haifeng Sun**[†]   **Menghao Zhang**
**Qi Qi**   **Zirui Zhuang**   **Jing Wang**   **Jianxin Liao**   **Jingyu Wang**[†]
State Key Laboratory of Networking and Switching Technology,
Beijing University of Posts and Telecommunications
{chengeng, rpf, jianxf, hfsun, zhangmenghao, qiqi8266, zhuangzirui,
wangjing, liaojx, wangjingyu}@bupt.edu.cn

## Abstract

Multi-view 3D human pose estimation (HPE) leverages complementary information across views to improve accuracy and robustness. Traditional methods rely on camera calibration to establish geometric correspondences, which is sensitive to calibration accuracy and lacks flexibility in dynamic settings. Calibration-free approaches address these limitations by learning adaptive view interactions, typically leveraging expressive and flexible continuous representations. However, as the multiview interaction relationship is learned entirely from data without constraint, they are vulnerable to noisy input, which can propagate, amplify and accumulate errors across all views, severely corrupting the final estimated pose. To mitigate this, we propose a novel framework that integrates a noise-resilient discrete prior into the continuous representation-based model. Specifically, we introduce the *Uni-Codebook*, a unified, compact, robust, and discrete representation complementary to continuous features, allowing the model to benefit from robustness to noise while preserving regression capability. Furthermore, we propose an attribute-preserving and complementarity-enhancing Discrete-Continuous Spatial Attention (DCSA) mechanism to facilitate interaction between discrete priors and continuous pose features. Extensive experiments on three representative datasets demonstrate that our approach outperforms both calibration-required and calibration-free methods, achieving state-of-the-art performance.

## 1 Introduction

3D human pose estimation aims to estimate the 3D locations of keypoints from images or videos. It is important because 3D human skeletons are widely used in scenarios such as action recognition [21, 25], human mesh recovery [8, 12], and robotics manipulation [7, 34]. Although monocular pose estimation offers greater convenience in usage, its performance is limited by depth ambiguity and occlusion issues, which hinder its broader application. Multiview systems [36, 13, 22, 31] offer a promising solution by leveraging complementary information from different views, enhancing accuracy and robustness in pose estimation. Existing multiview methods utilize the geometric constraints provided by camera extrinsics to establish correspondences between views, facilitating the fusion of complementary information to enhance 3D pose prediction. There are several drawbacks to using camera calibration-based methods. First, due to the complexity and computational intensity of camera

---

[*]Equal contribution.
[†]Corresponding authors.

39th Conference on Neural Information Processing Systems (NeurIPS 2025).

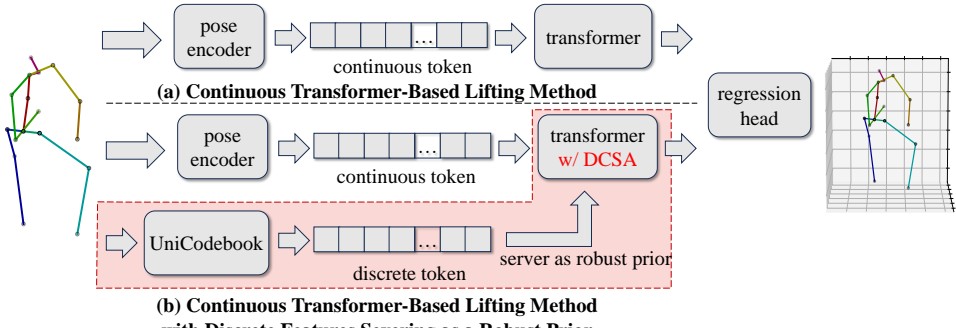

Figure 1: (a) Continuous transformer-based lifting method, which directly processes 2D pose inputs to estimate 3D poses. (b) Proposed method, which integrates discrete features as a robust prior within a continuous transformer-based framework, enhancing robustness to noisy 2D inputs and improving pose estimation accuracy.

calibration, these methods incur higher costs and exhibit limited flexibility. Second, their performance is sensitive to the precision of camera extrinsics, which can be unsatisfactory in complex or dynamic environments. Therefore, uncalibrated multi-view pose estimation has emerged as a significant trend, as it avoids the impact of unreliable camera extrinsics and is more flexible in deployment, leading to better generalization capability and broader applicability in real-world scenarios.

Many uncalibrated methods leverage adaptive multiview interaction patterns, relying on semantic intra-view and inter-view relationships to eliminate reliance on rigid geometric constraints. In recent years, purely transformer-based methods [32, 1, 2] have attracted increasing attention in calibration-free pose estimation, as they provide a unified and scaleable framework for modeling spatial, view, and temporal relationships using attention mechanisms. Notably, FusionFormer [1] and PoseIRM [2] have exhibited performance that surpasses even that of calibrated methods. This observation suggests that uncalibrated methods, rather than being constrained by predefined geometric relationships, can instead exploit data-driven adaptive semantic constraints. Such flexibility enables them to more effectively capture context-aware dependencies across views, thereby learning richer and more robust feature representations. For example, calibrated methods depend heavily on geometric information from external calibration, which occlusions can easily disrupt. In contrast, without calibration, uncalibrated methods are forced to leverage multi-view interactions to dynamically correct errors.

The superiority of transformer-based calibration-free framework can be attributed to their expressive and flexible continuous representations, which enable fine-grained modeling of pose dependencies when equipped with attention mechanism. Their continuity aligns with the continuous nature of both input and output spaces in 2D-to-3D pose lifting, enabling smooth, consistent, and precise mappings that are free from quantization errors and easy to optimize. However, this flexibility comes at the cost of increased susceptibility to overfitting when trained on limited data [5], making them sensitive to noisy 2D poses caused by occlusions and other common real-world challenges. Such errors can propagate, amplify and accumulate across views, affecting the accuracy of predictions from other views. Human pose is fundamentally constrained by inherent anatomical priors. These biological constraints, such as skeletal structure, joint motion limitations, and other biomechanical properties, define a natural boundary for valid poses. By encoding such anatomical knowledge as biomechanical priors in pose estimation models, we can intrinsically restrict the solution space to exclude implausible poses (*e.g.*, hyperextended elbows or inverted knees). Explicitly inject priors for guidance has been proven highly effective in other domains, such as image generation [44] and 3D reconstruction [40, 41]. This prior-based regularization complements data-driven approaches by injecting domain-specific knowledge, significantly improving robustness against noisy inputs, and is particularly valuable in uncalibrated scenarios where semantic constraints are entirely learned from the input data.

Despite the promise of integrating biomechanical prior-based regularization, effectively incorporating them into pose estimation models remains a non-trivial challenge. First, a core difficulty lies in how to represent such priors in a way that is structurally robust in representation. Discrete representations offer a promising solution by clustering large amount of body configurations into a limited set of proto-typical poses, which encourages the learning of high-quality dominant pose prototypes. This compression enforces structural regularity by restricting the model to select from a predefined pose vocabulary, thereby reducing overfitting to spurious or implausible patterns. Moreover, such many-to-one mappings, *i.e.*, multiple inputs can be mapped to the same prototype, naturally introduces tolerance to

noisy variations, as structurally similar but perturbed poses are treated equivalently. Although it is data-drive, such quantized space can serve as compact, interpretable, and noise-resilient priors-especially beneficial in ambiguous or weakly constrained estimation scenarios. Second, integrating discrete priors into continuous regression models introduces a representational mismatch. While discrete pose spaces are optimized for compactness and structural regularity, pose regressors demand fine-grained flexibility. Directly constraining outputs to lie in the discrete space may hinder expressiveness.

As illustrated in Figure 1, to address the challenges of constructing and leveraging human pose priors, we propose a unified, compact, and noise-resilient discrete prior-enhanced pipeline for 2D-to-3D human pose estimation. Central to our framework is the *UniCodebook*, which encodes both 2D and 3D poses as compositions of discrete sub-structure tokens within a shared discrete space to server as an informative prior that captures essential pose patterns and promotes resilience to noise. While discrete priors provide strong structural regularization and robustness to noise, their quantized nature inherently mismatches the continuous solution space required for accurate pose regression. Rather than directly regularizing the regressed output, we propose a Discrete-Continuous Spatial Attention (DCSA) module to enable continuous pose features to selectively attend to relevant discrete prototypes, which preserves the continuity of the original regression pipeline while softly integrating structural guidance from the discrete prior. By avoiding hard projection into the discrete space, DCSA mitigates representational mismatch and allows the model to inherit noise robustness from the discrete prior without sacrificing fine-grained precision, thereby effectively reconciling the gap between discrete structure and continuous prediction. We demonstrate the effectiveness by achieving state-of-the-art results on three benchmark datasets covering multiple complex scenarios.

The main contributions of the article are as follows:

- **VQ-VAE as a Robust Prior for Pose Estimation**: We propose the use of VQ-VAE to serve as a robust prior for pose estimation. By unifying 2D and 3D pose representations within a shared codebook, VQ-VAE further enhances its robustness to noisy pose, ensuring more accurate and consistent predictions, especially in challenging scenarios such as occlusion or 2D detection errors.
- **Discrete-Continuous Representation Interaction**: We introduce an attribute-preserving and complementarity-enhancing mechanism for the interaction between discrete (codebook-based) and continuous (transformer-based) representations. This allows the discrete tokens to guide the continuous features, infusing noise-resilient properties into the backbone while preserving the regression capability of the continuous representation in continuous tasks (*i.e.*, pose estimation).
- **State-of-the-Art Performance**: The proposed approach is capable of calibration-free human pose estimation even under significant noise, achieving SOTA performance on three large datasets.

## 2 Related Work

### 2.1 Multiview Calibration-Free 3D HPE

Methods with unknown camera extrinsics typically rely on semantic information rather than geometric information to enable better multiview feature interaction. FLEX [11] exploits the view-invariance characteristics of skeleton representations, such as bone lengths and rotation angles, to reconstruct human poses. MTF-TransFormer [32] and FusionFormer [1] leverage transformers to elegantly and efficiently model multiview, temporal, and spatial information, with the latter notably surpassing the performance of calibrated methods for the first time. PoseIRM [2] treats the uncalibrated task as a domain generalization problem and employs the invariant risk minimization paradigm to enhance performance in unseen camera settings. $A^3$-Net [3] leverages pre-defined alignment proxies, such as meshes and joints, to automatically discover inter-view interaction patterns.

### 2.2 VQ-VAE in Pose-Related Tasks

Recent advancements in pose-related tasks have leveraged VQ-VAE [37] to enhance representation learning and improve task performance. In PCT [10], they introduced a two-step framework for 2D pose estimation, where VQ-VAE is first used to learn a compact, discrete pose representation space, followed by a translator that maps image features to this space. This approach effectively prevents unrealistic poses and provides prior knowledge for occluded pose prediction. In TokenHMR [6], they applied a similar pipeline, focusing on learning pose parameters of the SMPL [23] body model using VQ-VAE. In MEGA [9], they utilized VQ-VAE to discretize 3D human meshes, achieving enhanced

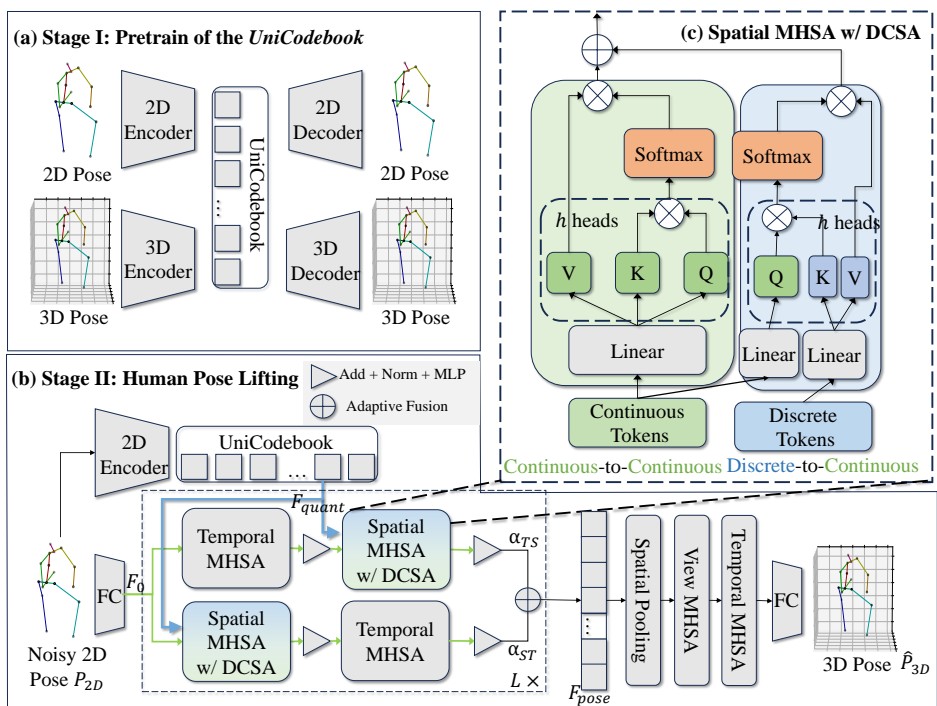

Figure 2: Two stages of the proposed calibration-free multiview 3D human pose lifting pipeline (a, b) and the detailed structure of the Spatial Multi-Head Self-Attention (MHSA) with Discrete-Continuous Spatial Attention (DCSA) (c). In Stage I, we construct the *UniCodebook*, a unified discrete representation space, through a multi-strategy training scheme (2Dto2D, 2Dto3D, 3Dto2D, and 3Dto3D). In this space, both 2D and 3D poses are encoded as sets of discrete tokens in this shared space to bridge the representation gap between 2D and 3D data. In Stage II, a transformer-based continuous model is employed for pose lifting, where codebook tokens generated from the *UniCodebook* are injected into the hybrid spatial attention block. Here, the proposed DCSA mechanism is integrated with conventional MHSA to facilitate effective fusion between the noise-resilient discrete priors and expressive continuous pose features, which enhances the robustness to noisy 2D input.

mesh recovery from single RGB images through both deterministic and stochastic generation modes, significantly improving uncertainty estimation and multi-output predictions. In DVQVAE [48], they proposed a decomposed VQ-VAE approach for generating realistic human grasps, where hand components are separately encoded to enhance interaction with objects. Di$^2$Pose [39] is a discrete diffusion model for occluded 3D pose estimation, which combines pose quantization with a discrete diffusion process to model 3D poses in latent space, leading to superior performance on occlusion handling and physical plausibility. These methods highlight the versatility of VQ-VAE, demonstrating its potential in improving robustness and generalization across a wide range of pose related problems. However, these methods either rely on hard mappings between continuous features and discrete representations [9], or constrain the entire prediction or generation process to discrete latent spaces [48, 39, 10, 6], where quantization can obscure fine-grained details. In contrast, continuous representations excel at capturing subtle spatial variations, which are critical for precise and smooth pose reconstruction. To fully exploit the complementary strengths of discrete structure and continuous precision, our method softly integrates discrete structural priors into the continuous regression stream, enabling noise-robust yet precise pose prediction.

# 3 Methodology

Figure 2 depicts the overall architecture of the proposed methods. In Section 3.1, we describe how to learn an *UniCodebook* which represents 2D and 3D data simultaneously. Section 3.2 explains how the pretrained the *UniCodebook* is used in the human pose lifting task.

## 3.1 Stage I: Pretrain of the *UniCodebook*

In this section, we first introduce the key components used in Stage I. We begin with the Pose Encoder, which processes the original pose into several tokens, followed by the proposed *UniCodebook*, which retrieves features from this unified compact feature space. Finally, we introduce the Pose Decoder and the loss function used to train the entire pipeline.

Existing codebook-based pose representation methods suffer from **modality isolation**, *i.e.*, they encode either 2D poses [10] or 3D poses [6] in separate models. This isolation creates two critical issues: (1) underutilization of heterogeneous pose data that inherently contains complementary 2D-3D correlations, and (2) failure to align the representation gap between 2D observation and 3D geometry. Since the lifting task involves both 2D and 3D features, this gap forces the backbone model to allocate additional capacity to transform unimodal features into a more suitable representation for lifting, rather than focusing entirely on learning robust 2D-to-3D mappings.

To more effectively utilize 2D and 3D data and bridge the gap between them, we propose a unified cross-modal representation learning framework that establishes shared latent embeddings between 2D and 3D human poses. Specifically, while maintaining dedicated modality-specific encoders and decoders for 2D/3D data processing, we use a shared codebook, *i.e.*, the *UniCodebook*, that jointly represents both 2D and 3D pose. We optimize the whole framework through four complementary objectives: 2Dto2D self-reconstruction, 3Dto3D self-reconstruction, cross-modal 2Dto3D projection, and inverse 3Dto2D back-projection. The first two auto-encoding tasks preserve modality-specific reconstruction fidelity and the latter two projection objectives enforce a smaller representation gap through bidirectional translation between modalities.

**Pose Encoder.** Given a human pose $\mathbf{P} \in \mathbb{R}^{J \times D}$ where $D = 2$ for 2D pose $\mathbf{P}_{2d}$ and $D = 3$ for 3D pose $\mathbf{P}_{3d}$, we adopt the **compositional encoding strategy** to capture human priors and structural relationships following [10]. Unlike joint-wise encoding that processes each joint independently, we decompose the pose into $N$ overlapping sub-structures $\mathbf{f}_i$ (e.g., limbs, torso combinations), where each sub-structure corresponds to a group of biomechanically correlated joints. It is noted that the compositional representation contains lots of redundancy due to the fact that different tokens may share overlapping joints. The redundancy, in turn, enhances the robustness of the representation against occlusions of individual parts [10].

We utilize two separate encoders, both identical in structure, to encode 2D and 3D poses, respectively. Following the approach in [10], we adopt the MLP-Mixer architecture [35] as our encoder $f_e$, which is particularly adept at capturing both local joint interactions and global pose semantics. Given an input pose $\mathbf{P}$, we first embed it into $\mathbf{P}_{\text{emb}} \in \mathbb{R}^{J \times D_{\text{feat}}}$ using a linear layer $f_{\text{emb}}$. Subsequently, the embedded features are passed through four MLP-Mixer blocks to effectively model the relationships between different joints. Finally, we employ another linear layer $f_{\text{trans}}$ to map the features to $N$ tokens. The overall process is described by $\mathbf{F}_{\text{en}} = [\mathbf{f}_1, \mathbf{f}_2, \ldots, \mathbf{f}_N] = f_{\text{trans}}(f_e(f_{\text{emb}}(\mathbf{P}))) \in \mathbb{R}^{N \times d}$, where each $\mathbf{f}_i \in \mathbb{R}^{D_{feat}}$. The number of tokens is set to $N = 63$, which is larger than the 51 input dimensions (17 joints × 3) of a Human3.6M pose, to introduce representational redundancy.

**Quantization Process With the *UniCodebook*.** We define a shared *UniCodebook* space $\mathbf{C} = [\mathbf{c}_1, \ldots, \mathbf{c}_U]^{\top} \in \mathbb{R}^{U \times D_{\text{feat}}}$ that jointly represents both 2D and 3D poses. Specifically, each codebook entry $\mathbf{c}_u$ encodes pose patterns applicable to both modalities. This shared representation of discrete tokens helps to minimize the disparity between heterogeneous pose representations.

For each sub-structure token $\mathbf{f}_i$, we perform quantization via nearest neighbor lookup in the codebook:

$$q(\mathbf{f}_i = u \mid \mathbf{P}) = \begin{cases} 1 & \text{if } u = \arg\min_j \|\mathbf{f}_i - \mathbf{c}_j\|_2 \\ 0 & \text{otherwise} \end{cases} \tag{1}$$

We abuse $q(\mathbf{f}_i)$ to denote the index of the closest codebook entry for token $\mathbf{f}_i$. The quantized pose representation is given by $\mathbf{F}_{\text{quant}} = [\mathbf{c}_{q(\mathbf{f}_1)}, \mathbf{c}_{q(\mathbf{f}_2)}, \ldots, \mathbf{c}_{q(\mathbf{f}_N)}] \in \mathbb{R}^{N \times D_{\text{feat}}}$.

**Pose Decoder.** The pose decoder is designed to recover pose $\hat{\mathbf{P}}$ from quantized feature $\mathbf{F}_{\text{quant}}$. It mirrors the architecture of the pose encoder but in a reverse manner.

**Loss.** For each strategy, we optimize the entire framework using the following loss function:

$$\mathcal{L}_{\text{pretrain}} = \text{smooth}_{\mathcal{L}_1}(\hat{\mathbf{P}}, \mathbf{P}) + \beta \sum_{i=1}^{M} \|\mathbf{f}_i - \text{sg}[\mathbf{c}_{q(\mathbf{f}_i)}]\|_2^2 \tag{2}$$

where sg denotes stopping gradient, and $\beta$ is a hyperparameter which we set to 10.

## 3.2  Stage II: Human Pose Lifting

Our framework consists of three main components at Stage II: (1) 2D pose discretization and initial continuous feature embedding, (2) dual-stream spatial-temporal feature interaction, and (3) cross-view and temporal feature refinement, and 3D pose regression.

Given a 2D input skeleton sequence $\mathbf{P}_{2D} \in \mathbb{R}^{V \times T \times J \times D_{2D}}$, we first feed it into the 2D Pose Encoder of the VQ-VAE trained in Stage I and quantize it into discrete codebook features $\mathbf{F}_{quant} \in \mathbb{R}^{V \times T \times N \times D_{feat}}$. Subsequently, within the transformer backbone, we initially project the 2D pose $\mathbf{P}_{2D}$ into a higher-dimensional feature space $\mathbf{F}^0 \in \mathbb{R}^{V \times T \times J \times D_{feat}}$ via a fully-connected (FC) layer. We then employ a dual-stream transformer equipped with spatial and temporal Multi-Head Self-Attention (MHSA) mechanisms to derive the feature $\mathbf{F}_{pose} \in \mathbb{R}^{V \times T \times J \times D_{feat}}$. Within the spatial MHSA, we implement Discrete-Continuous Spatial Attention (DCSA) to facilitate interaction between continuous features and the discrete features $\mathbf{F}_{quant}$ along the spatial dimension.

Next, we apply spatial pooling to obtain $\mathbf{F}_{pool} = \text{AveragePooling}(\mathbf{F}_{pose}) \in \mathbb{R}^{V \times T \times D_{feat}}$. This is followed by the application of View MHSA and Temporal MHSA to yield the final continuous pose features $\mathbf{F}'_{pose} \in \mathbb{R}^{V \times T \times D_{feat}}$. Finally, we regress the predicted 3D pose $\hat{\mathbf{P}}_{3D} \in \mathbb{R}^{V \times T \times J \times D_{3D}}$ through an FC layer. We only apply L1 loss between ground truth(GT) 3D pose $\mathbf{P}_{3D}$ and $\hat{\mathbf{P}}_{3D}$. For simplicity, in the following descriptions, we omit layer notations unless cross-layer feature interaction is involved.

### 3.2.1  Hybrid Spatial Attention Block

The spatial block contains a Spatial Multihead Self-Attention (MHSA) and a Discrete-Continuous Spatial Attention (DCSA) module. For simplicity, we describe the computation for a single attention head. The multi-head version follows standard practice by computing heads in parallel and projecting the concatenated results.

**Spatial MHSA** models relationships between joints within the same view and frame. Given a spatial feature $\mathbf{F}_S \in \mathbb{R}^{J \times D_{feat}}$ from $\mathbf{F}_{pose}$, we first compute its query, key, and value vectors:

$$\mathbf{Q}_S = \mathbf{F}_S \mathbf{W}_S^Q, \quad \mathbf{K}_S = \mathbf{F}_S \mathbf{W}_S^K, \quad \mathbf{V}_S = \mathbf{F}_S \mathbf{W}_S^V, \tag{3}$$

where $\mathbf{W}_S^Q, \mathbf{W}_S^K, \mathbf{W}_S^V \in \mathbb{R}^{D_{feat} \times D_K}$ are learnable projection matrices. The self-attention output is then computed as:

$$\text{S-MHSA}(\mathbf{F}_S) = \text{softmax}\left(\frac{\mathbf{Q}_S \mathbf{K}_S^\top}{\sqrt{D_K}}\right) \mathbf{V}_S, \tag{4}$$

where $D_K$ is the dimension of the key vectors. The temporal MHSA follows the same computation but applies to temporal tokens.

**Discrete-Continuous Spatial Attention (DCSA)** augments the continuous joint interactions with semantic guidance from discrete codebook tokens. We implement this module using a cross-attention mechanism. As detailed in our ablation study in Table 4b and Table 7 in the Supplementary Materials, this approach proved to be the most effective and straightforward compared to other alternatives. The mechanism derives queries from the continuous features $\mathbf{F}_S$ and keys/values from the discrete codebook tokens $\mathbf{F}_{quant} \in \mathbb{R}^{N \times D_{feat}}$:

$$\mathbf{Q}_D = \mathbf{F}_S \mathbf{W}_D^Q, \quad \mathbf{K}_D = \mathbf{F}_{quant} \mathbf{W}_D^K, \quad \mathbf{V}_D = \mathbf{F}_{quant} \mathbf{W}_D^V. \tag{5}$$

The discrete-to-continuous attention is then computed as:

$$\text{DCSA}(\mathbf{F}_S, \mathbf{F}_{quant}) = \text{softmax}\left(\frac{\mathbf{Q}_D \mathbf{K}_D^\top}{\sqrt{D_K}}\right) \mathbf{V}_D. \tag{6}$$

The final spatial representation combines features from both modules through addition:

$$\mathbf{F}_S^{out} = \underbrace{\text{S-MHSA}(\mathbf{F}_S)}_{\text{Continuous-to-Continuous}} + \underbrace{\text{DCSA}(\mathbf{F}_S, \mathbf{F}_{quant})}_{\text{Discrete-to-Continuous}}. \tag{7}$$

The output features undergo residual connections and layer normalization (LayerNorm), followed by an multilayer perceptron (MLP) block with another residual connection and LayerNorm. We denote the entire spatial block by $\mathcal{S}$.

### 3.2.2 Dual-Stream Spatial-Temporal Transformer

Following MotionBert [49], we construct a dual-stream spatial-temporal transformer by alternately stacking temporal and hybrid spatial blocks in different order, thereby creating two parallel branches.

To enable the model to dynamically adjust the importance of different branches, we compute an adaptive weight for each branch. Adaptive fusion weights $\alpha_{ST}^i, \alpha_{TS}^i \in \mathbb{R}^{L \times V \times T \times J}$ are predicted by linear projection, as $\alpha_{ST}^i, \alpha_{TS}^i = \text{softmax}(\mathcal{W}([\mathcal{T}_1^i(\mathcal{S}_1^i(\mathbf{F}_{\text{pose}}^{i-1})), \mathcal{S}_2^i(\mathcal{T}_2^i(\mathbf{F}_{\text{pose}}^{i-1}))]))$, where $\mathcal{W}$ is a learnable linear transformation, $[\cdot, \cdot]$ denotes concatenation, $i \in \{1, \ldots, L\}$ is the current layer number, and $L$ is the number of layers of the dual-stream module.

The features from the two branches are fused additively via predicted weights as $\mathbf{F}_{\text{pose}}^i = \alpha_{ST}^i \circ \mathcal{T}_1^i(\mathcal{S}_1^i(\mathbf{F}_{\text{pose}}^{i-1})) + \alpha_{TS}^i \circ \mathcal{S}_2^i(\mathcal{T}_2^i(\mathbf{F}_{\text{pose}}^{i-1}))$, where $\circ$ denotes element-wise multiplication.

## 4 Experiments

### 4.1 Datasets and Metrics

**Human3.6M** [14] is a widely used 3D human pose dataset with 3.6 million frames captured by 4 synchronized cameras, covering 15 actions performed by 11 subjects. And the poses are represented by 17 human joints. Following common protocol, we use subjects S1, S5, S6, S7, and S8 for training, and S9 and S11 for testing. We evaluate under two settings: using ground-truth (GT) 2D poses and CPN-detected 2D poses. The CPN detector introduces significant 2D pose noise: the average 2D MPJPE between CPN-detected and ground-truth keypoints is 3.9 pixels on the training set and 6.5 pixels on the test set, with maximum errors reaching 55 and 124 pixels, respectively.

**MPI-INF-3DHP** [27] is a challenging multiview dataset for 3D human pose estimation, consisting of 1.3 million frames performed by 8 subjects in diverse indoor and outdoor scenarios, which is widely used to evaluate the generalization performance in complex real-world scenarios. Following previous studies [20, 16], we train on subjects S1-S7 and test on subject S8.

**FreeMan** [38] is a large-scale multiview 3D human pose dataset collected under real-world conditions, containing 11 million frames from 8,000 sequences captured by 8 synchronized smartphones. It covers 40 subjects across 10 diverse indoor and outdoor scenarios with varying lighting. To address the limited distribution gap between the training and testing sets in the original protocol, we re-split the dataset that introduces clearer separation in both scene and camera configurations: scenes 1–8 and cameras (1,3,5,7) are used for training, while scenes 10–12 and cameras (1,3,5,7) (2,4,6,8) are reserved for testing. The resulting training set consists of 566,484 image pairs, and the test set includes 646,608 pairs. This revised split enables a more realistic and challenging evaluation of model generalization.

**AMASS** [26] is a large-scale motion capture dataset with over 40 hours of motion data from 300+ subjects and 11,000+ motions. Since it does not provide direct 2D-3D annotations, we render SMPL meshes and apply official Human3.6M and COCO regressors to extract 3D joints, which are then projected onto 2D planes from 8 virtual views to form 2D-3D pairs. We use the official validation and test splits for training and evaluating the VQ-VAE in Stage I only.

**Evaluation Metrics.** We report Mean Per Joint Position Error (MPJPE), measured in millimeters, Percentage of Correct Keypoints (PCK), and Area Under Curve (AUC) with a 150mm threshold.

### 4.2 Implementation Details

All individual experiments were completed within 1 day using one server equipped with an Nvidia RTX 3090 GPU, 64GB RAM, and a 28-core CPU. We employed the AdamW [24] optimizer with a learning rate of 2e-4 for both the Stage I and Stage II. We used a cosine annealing learning scheduler with 100 warmup steps. The model was trained for 50 and 100 epochs with a batch size of 256 at Stage I and Stage II, respectively. No data augmentation techniques were applied during training and test. For VQ-VAE, we use $N = 63$ tokens to represent a pose, and the codebook is in shape $U \times D_{\text{feat}} = 2048 \times 256$. The original VQ-VAE suffers from codebook collapse, where a significant portion of codebook entries remain underutilized. Following [6], we employ exponential moving average (EMA) and code reset tricks to enhance codebook activation rates.

Table 1: Results on Human3.6M are reported using MPJPE as the evaluation metric. CPN, HRNet and ResNet152 are different 2D pose detectors. GT means using ground truth 2D pose. * means this is an image-to-3d method. † indicates our reimplementation. T represents the number of frames.

| Method | Venue | Input Setting | Dir. | Disc. | Eat. | Greet | Phone | Photo | Pose | Purch. | Sit. | SitD. | Smoke | Wait | WalkD. | Walk | WalkT. | Avg. |
|---|---|---|---|---|---|---|---|---|---|---|---|---|---|---|---|---|---|---|
| Monocular Methods | | | | | | | | | | | | | | | | | | |
| MHFormer [19] | CVPR'2022 | (CPN, T = 351) | 39.2 | 43.1 | 40.1 | 40.9 | 44.9 | 51.2 | 40.6 | 41.3 | 53.5 | 60.3 | 43.7 | 41.1 | 43.8 | 29.8 | 30.6 | 43.0 |
| MixSTE [43] | CVPR'2022 | (CPN, T = 243) | 37.6 | 40.9 | 37.3 | 39.7 | 42.3 | 49.9 | 40.1 | 39.8 | 51.7 | 55.0 | 42.1 | 39.8 | 41.0 | 27.9 | 27.9 | 40.9 |
| KTPFormer [29] | CVPR'2024 | (CPN, T = 243) | 30.1 | 32.1 | 29.1 | 30.6 | 35.4 | 39.3 | 32.8 | 30.9 | 43.1 | 45.5 | 34.7 | 33.2 | 32.7 | 22.1 | 23.0 | 33.0 |
| Multi-View Methods With Camera Parameter | | | | | | | | | | | | | | | | | | |
| Epipolar Transformer [13] | CVPR'2020 | (*, T = 1) | 25.7 | 27.7 | 23.7 | 24.8 | 26.9 | 31.4 | 24.9 | 26.5 | 28.8 | 31.7 | 28.2 | 26.4 | 23.6 | 28.3 | 23.5 | 26.9 |
| Crossview Fusion [30] | ICCV'2019 | (*, T = 1) | 24.0 | 26.7 | 23.2 | 24.3 | 24.8 | 22.8 | 24.1 | 28.6 | 32.1 | 26.9 | 31.0 | 25.6 | 25.0 | 28.0 | 24.4 | 26.2 |
| Geometry-Biased Transformer [28] | FG'2024 | (HRNet, T=27) | - | - | - | - | - | - | - | - | - | - | - | - | - | - | - | - |
| MTF-Transformer+ [32] | TPAMI'2022 | (CPN, T = 27) | 23.4 | 25.2 | 23.1 | 24.4 | 27.4 | 28.5 | 22.8 | 25.2 | 28.7 | 36.2 | 25.9 | 23.6 | 26.6 | 22.6 | 22.7 | 25.8 |
| LearnTriangulation [15] | ICCV'2019 | (*, T = 1) | 19.9 | 20.0 | 18.9 | 18.5 | 20.5 | 19.4 | 18.4 | 22.1 | 22.5 | 28.7 | 21.2 | 20.8 | 19.7 | 22.1 | 20.2 | 20.8 |
| AdaFuse [47] | IJCV'2021 | (*, T = 1) | 17.8 | 19.5 | 17.6 | 20.7 | 19.3 | 16.8 | 18.9 | 20.2 | 25.7 | 20.1 | 19.2 | 20.5 | 17.2 | 20.5 | 17.3 | 19.5 |
| MvP [42] | NeurIPS'2021 | (*, T = 1) | - | - | - | - | - | - | - | - | - | - | - | - | - | - | - | 18.6 |
| Multi-View Methods Without Camera Parameter | | | | | | | | | | | | | | | | | | |
| ProTriangulation [17] | ICCV'2023 | (*, T = 1) | 24.0 | 25.4 | 26.6 | 30.4 | 32.1 | 20.1 | 20.5 | 36.5 | 40.1 | 29.5 | 27.4 | 27.6 | 20.8 | 24.1 | 22.0 | 27.8 |
| FLEX [11] | ECCV'2022 | (ResNet152, T = 27) | 23.1 | 28.8 | 26.8 | 28.1 | 31.6 | 37.1 | 25.7 | 31.4 | 36.5 | 39.6 | 35.0 | 29.5 | 35.6 | 26.8 | 26.4 | 30.9 |
| FLEX [11] | ECCV'2022 | (CPN, T = 27) | - | - | - | - | - | - | - | - | - | - | - | - | - | - | - | 31.7 |
| SGraFormer [45] | AAAI'2024 | (CPN, T = 27) | 26.5 | 28.3 | 23.0 | 25.9 | 27.2 | 31.0 | 25.4 | 27.2 | 28.6 | 33.8 | 28.6 | 25.6 | 30.1 | 27.1 | 26.5 | 27.6 |
| FusionFormer [1] † | AAAI'2024 | (CPN, T = 27) | 23.7 | 26.3 | 24.8 | 25.3 | 27.7 | 27.1 | 23.9 | 26.6 | 32.3 | 32.7 | 25.9 | 25.2 | 27.9 | 23.5 | 23.8 | 26.6 |
| Ours | | (CPN, T = 27) | 23.0 | 25.2 | 24.8 | 24.4 | 27.5 | 26.4 | 23.4 | 25.6 | 31.4 | 32.3 | 25.7 | 24.0 | 27.3 | 22.9 | 22.9 | 26.0 |
| MTF-Transfomer [32] | TPAMI'2022 | (GT, T = 27) | 15.5 | 17.1 | 13.7 | 15.5 | 14.0 | 16.2 | 15.8 | 16.5 | 15.8 | 16.1 | 14.5 | 14.5 | 16.9 | 14.3 | 13.7 | 15.3 |
| SGraFormer [45] | AAAI'2024 | (GT, T = 27) | 11.7 | 13.0 | 10.1 | 12.1 | 10.7 | 13.0 | 12.1 | 10.7 | 10.8 | 11.9 | 11.0 | 11.6 | 12.8 | 11.1 | 12.0 | 11.7 |
| SVTFormer [46] | AAAI'2025 | (GT, T = 27) | 11.6 | 12.3 | 11.4 | 12.2 | 10.6 | 12.1 | 12.5 | 11.7 | 10.3 | 10.7 | 10.7 | 12.1 | 10.8 | 10.9 | 11.4 | 11.4 |
| FusionFormer [1] | AAAI'2024 | (GT, T = 27) | 7.84 | 8.04 | 7.39 | 8.33 | 7.13 | 9.02 | 8.00 | 8.19 | 7.57 | - | 7.37 | 7.83 | - | 7.26 | - | 7.90 |
| Ours | | (GT, T = 27) | 7.77 | 7.82 | 7.70 | 7.71 | 7.80 | 7.76 | 7.87 | 7.60 | 7.81 | 7.66 | 7.76 | 7.92 | 7.63 | 7.66 | 7.74 | 7.74 |

Table 2: Comparison results on the MPI-INF-3DHP.

| Method | Extra Data for Finetune | MPJPE↓ | PCK↑ | AUC↑ |
|---|---|---|---|---|
| ShapeAware [20] | syn data | 62 | 95 | 63 |
| PaFF [16] | × | 48.4 | 98.6 | 67.3 |
| SGraFormer [45] | × | 16.9 | 98.7 | 90.2 |
| MTF-Transformer [32] | h36m | 14.6 | - | - |
| SGraFormer [45] | h36m | 10.6 | 99.9 | 91.7 |
| FusionFormer [1] | h36m | 5.4 | - | - |
| Ours | × | 3.37 | 99.9 | 95.94 |

Table 3: Comparison results on the FreeMan dataset. † indicates our reimplementation.

| | seen camera MPJPE↓ | unseen camera MPJPE↓ | mean MPJPE↓ |
|---|---|---|---|
| SGraFormer [45] † | 22.20 | 28.69 | 25.44 |
| MTF [32] † | 10.85 | 13.51 | 12.18 |
| Fusionformer [1] † | 9.89 | 12.72 | 11.30 |
| Ours | 9.59 | 12.71 | 11.15 |

## 4.3 Comparisons With SOTAs

**Human 3.6M.** Table 1 compares our method with several state-of-the-art (SOTA) monocular and multi-view approaches on the Human 3.6M dataset for 3D human pose estimation (HPE). Our method outperforms all the calibrated methods due to that our method are not limited by predefined geometric relationships between views but instead utilizes data-driven adaptive semantic constraints, allowing for more effective multi-view feature fusion. When compared to calibration-free 3D HPE methods using CPN [4] or GT 2D poses, our approach achieves SOTA performance, with MPJPE values of **26.0mm** and **7.74mm**, respectively, demonstrating its effectiveness. Notably, our method does not rely on any additional tricks but simply adopts L1 Loss.

**MPI-INF-3DHP.** Table 2 presents a detailed comparison with other SOTA calibration-free methods on the MPI-INF-3DHP dataset. We use GT 2D poses as input and keep the same settings as [20]. As shown in the table, our method consistently outperforms all other methods across various metrics. Specifically, we achieve a significant reduction in the MPJPE, outperforming the second-best method by at least **2.03mm (37.5%)**. Additionally, our method shows a notable increase in the AUC, improving it by at least **4.24%**. These indicate that our model not only achieves higher accuracy but also maintains robust performance across different thresholds. It also demonstrates that the *UniCodebook* not only enhances robustness but also improves accuracy without noise.

**FreeMan.** Table 3 reports the evaluation results on the FreeMan dataset under seen and unseen camera settings using HRNet-w48 [33] 2D poses. Our method achieves the best overall performance with a mean MPJPE of 11.15mm, outperforming recent approaches such as Fusionformer [1] (11.30mm) and MTF [32] (12.18mm). In particular, our method attains the lowest error under unseen camera settings (9.59mm), demonstrating strong generalization to novel viewpoints. These results confirm the robustness and adaptability of our approach in real-world scenarios with cross-camera configurations.

## 4.4 Ablation Study

**Impact of Codebook Training Strategy.** Table 4a shows how different codebook training strategies in Stage I affect the lifting performance at Stage II. The best performance is achieved when all strategies are used. Without the codebook, performance drops significantly, highlighting its importance as a robust prior for the continuous model. The MPJPE are around 26.8, 26.7, 26.4 and 26.3 when

Table 4: Ablation studies on key components of the *UniCodebook*, including training strategies, feature interaction methods, and data sources. MPJPE is calculated at Stage II. "CPN" and "GT" denote 2D poses from CPN [4] and ground truth, respectively.

(a) Ablation study on different training strategy combinations for the *UniCodebook* where T=1.

| Number of Strategy | Strategy | | | | MPJPE↓ |
|---|---|---|---|---|---|
| | 2dto2d | 2dto3d | 3dto2d | 3dto3d | |
| 0 | | | | | 27.01 |
| 1 | ✓ | | | | 26.74 |
| | | ✓ | | | 26.86 |
| 2 | ✓ | ✓ | | | 26.70 |
| | ✓ | | ✓ | | 26.78 |
| | ✓ | | | ✓ | 26.74 |
| | | ✓ | ✓ | | 26.73 |
| | | ✓ | | ✓ | 26.75 |
| 3 | ✓ | ✓ | ✓ | | 26.40 |
| | ✓ | ✓ | | ✓ | 26.47 |
| | ✓ | | ✓ | ✓ | 26.53 |
| | | ✓ | ✓ | ✓ | 26.45 |
| 4 | ✓ | ✓ | ✓ | ✓ | **26.34** |

(b) Ablation experiments on different interaction methods between discrete codebook features and continuous features. T=27.

| Method | MPJPE↓ |
|---|---|
| Baseline | 26.7 |
| Conv + Addition | 26.6 |
| Q-Former [18] | 34.1 |
| Concat + MHSA | 26.4 |
| DCSA | **26.0** |

(c) Ablation on training data of VQ-VAE. T=27.

| Data Source | MPJPE↓ |
|---|---|
| H36M CPN | 26.43 |
| H36M GT | 26.35 |
| AMASS | **26.02** |

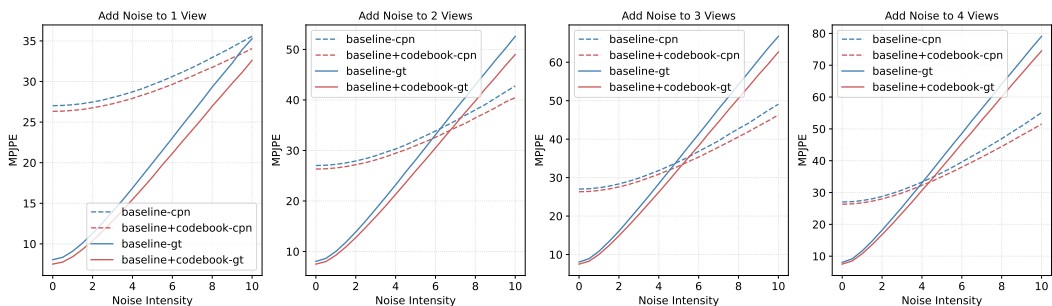

Figure 3: Comparison of MPJPE error across four models (*i.e.*, baseline trained on H36M CPN, baseline with codebook trained on H36M CPN, baseline trained on H36M GT, and baseline with codebook trained on H36M GT) under varying noise intensities without retraining. For each instance (consisting of multi-view 2D poses of the same person at the same timestamp), we randomly select 1 to 4 views and add Gaussian noise with zero mean and a standard deviation of "Noise Intensity" pixels to each 2D joint. For models trained on H36M CPN, we evaluate them using H36M CPN test data with extra noise. Similarly, models trained on H36M GT are evaluated with H36M GT test data with extra noise. **The results show that models with the codebook exhibit robustness across all noise levels, with greater robustness observed at higher noise intensities.**

applying one, two, three and four strategies, respectively. This decreasing trend demonstrates that as more strategies are used, the *UniCodebook* can better capture the relationships between 2D and 3D poses and data utilization rate , making it a robuster prior for lifting tasks.

**Impact of the Interaction Method of Discrete Codebook Features and Continuous Features.** In Table 4b, we aim to investigate the impact of different interaction methods between discrete and continuous features on model performance. "Baseline" means no discrete features are used. "Conv + Addition" involves aligning the token count of the codebook features using a convolutional layer, followed by adding them to the continuous features. "Q-Former [18]" employs learned queries with the same shape as continuous features and maps discrete feature information onto the queries using a transformer layer. The queries and continuous features then interact via cross-attention. "Concat + MHSA" first concatenates discrete and continuous features along the token dimension and then applies self-attention. All methods are applied in multiple spatial blocks. The results demonstrate that our DCSA method achieves the best performance, improving MPJPE by 0.7mm.

**Impact of Noise.** To evaluate the robustness of our model to noisy inputs, we present in Figure 3 the MPJPE performance of four models under varying noise intensities. The results demonstrate that

Table 5: Ablation on the number of pose tokens and codebook size in UniCodebook where the frame number is 1. The active rate represents the mean activation rate in Stage I, while MPJPE is evaluated in Stage II on Human3.6M [14].

| pose tokens | codebook size | Active Rate(%)↑ | MPJPE↓ |
|---|---|---|---|
| 21 | | 61.8 | 26.82 |
| 63 | 2048 | 83.5 | **26.34** |
| 105 | | **84.3** | 26.35 |
| | 1024 | **84.5** | 26.70 |
| 63 | 2048 | 83.5 | **26.34** |
| | 4096 | 75.0 | 26.47 |

our method—whether trained on H36M CPN or H36M GT—consistently outperforms the baseline in the presence of noise. Moreover, the performance gap widens as the noise intensity increases, indicating that the codebook-based model exhibits stronger resilience to noisy 2D observations than the continuous transformer-based baseline. This robustness stems from the discrete nature of the codebook, which provides a form of structural regularization by anchoring predictions to a set of stable pose prototypes. Such discrete priors may help correct noisy or ambiguous joint inputs by guiding the model toward plausible configurations. These findings highlight the effectiveness of our approach in real-world scenarios, where noisy 2D detections are common, and underscore the practical advantages of incorporating discrete structural knowledge into continuous pose estimation models.

**Impact of Codebook Training Data Quality.** To evaluate the impact of data quality on model performance, we conducted experiments in Table 4c using three data sources: H36M CPN, H36M GT, and AMASS. The results show a consistent performance increase among them. As data quality improves, the model performance also increases, indicating that higher-quality data leads to better performance. It is worth noting that even when using only H36M CPN data, *i.e.*, without any external data, our model still achieves near-SOTA performance on the Human3.6M benchmark.

**Impact of Codebook Configurations.** We analyze two key hyperparameters: the **codebook size**, which defines the capacity of the discrete latent space, and the **number of pose tokens**, which reflects the granularity of the pose representation. (1) First, we fixed the codebook size to 2048 and varied the number of pose tokens. Increasing the tokens from 21 to 63 significantly boosted the active rate (from 61.8% to 83.5%) and improved MPJPE (from 26.82 to 26.34). A further increase to 105 tokens yielded no improvement, suggesting that 63 tokens offer a favorable balance between expressiveness and efficiency. (2) Next, we fixed the number of pose tokens at 63 and varied the codebook size. A codebook of 2048 achieved the best MPJPE (26.34). A smaller size (1024) suffered from higher error (MPJPE: 26.70) due to limited capacity, while a larger one (4096) slightly degraded performance (MPJPE: 26.47). Overall, our selected configuration of **63 pose tokens** and a **codebook size of 2048** achieves the best trade-off between token utilization and pose estimation accuracy.

## 5 Conclusion

In this paper, we propose a novel approach for uncalibrated multi-view 3D human pose estimation by integrating a noise-resilient prior, *i.e.*, the *UniCodebook*, into a transformer-based framework. By employing a multi-strategy training scheme, the *UniCodebook* effectively bridges the gap between 2D and 3D pose representations. Furthermore, we introduce Discrete-Continuous Spatial Attention (DCSA) to facilitate interaction between discrete codebook tokens and continuous pose features, ensuring the benefits of noise-resilient priors without disrupting the continuity of the backbone model. Extensive experiments on three benchmark datasets demonstrate that our approach outperforms both calibration-required and calibration-free methods, achieving state-of-the-art performance. These results highlight the effectiveness of integrating discrete priors into multi-view pose estimation, demonstrating their potential to enhance the robustness of continuous models.

## Acknowledgement

This work was supported in part by the National Natural Science Foundation of China under Grants (62406039, 62321001, 62471055, U23B2001, 62171057, 62201072, 62071067, 62406039, 62101064), the High-Quality Development Project of the MIIT (2440STCZB2584), the Ministry of Education and China Mobile Joint Fund (MCM20200202, MCM20180101), the Project funded by China Postdoctoral Science Foundation (2023TQ0039, 2024M750257, GZC20230320), the Fundamental Research Funds for the Central Universities (2024PTB-004), and the 2025 Education and Teaching Reform Project Funding at Beijing University of Posts and Telecommunications (2025YZ005).

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
