# OpenReview forum: "Unified 2D-3D Discrete Priors for Noise-Robust and Calibration-Free Multiview 3D Human Pose Estimation"
_NeurIPS.cc/2025/Conference — NeurIPS 2025 poster_

### Official Review · Reviewer_63UD · 2025-06-18

**Clarity:** 3
**Significance:** 2
**Originality:** 2
**Rating:** 4
**Confidence:** 3

**Summary:**

This paper addresses the challenge of multi-view 3D human pose estimation (HPE), where traditional methods rely on precise camera calibration—limiting flexibility in dynamic settings—while calibration-free approaches, though more adaptable, suffer from error propagation due to unconstrained learning of view interactions. To overcome these limitations, the authors propose a novel framework that combines a noise-resilient discrete prior (Uni-Codebook) with continuous feature representations, enhancing robustness without sacrificing regression accuracy. Additionally, they introduce a Discrete-Continuous Spatial Attention (DCSA) mechanism to improve interaction between discrete and continuous features while preserving key attributes. Experiments on three benchmark datasets demonstrate that the proposed method outperforms both calibration-dependent and calibration-free approaches, achieving state-of-the-art performance in multi-view 3D HPE.

**Questions:**

1. What are the differences in experimental settings between the first row of Table 4(a) and the baseline in Table 4(b) that lead to the performance discrepancy?

2. The title format of Reference 32 is incorrect.

**Ethical Concerns:**

["NO or VERY MINOR ethics concerns only"]

**Final Justification:**

I sincerely apologize for not replying sooner. I have carefully reviewed the author's response as well as the discussions from the other reviewers. I greatly appreciate the author's detailed reply, which has resolved all of my concerns. Therefore, I have decided to increase my original score by one point.

**Limitations:**

The authors should discuss how the inference efficiency compares with FusionFormer.

**Quality:**

2

**Strengths And Weaknesses:**

**Strengths:**
1. The paper is clearly written, with a logical structure and well-organized presentation.
2. Comprehensive experimental evaluations are conducted across multiple datasets, demonstrating the generalizability of the proposed method.

**Major Weaknesses:**
1. Limited methodological novelty: This work proposes an incremental enhancement to existing methods by adding quantized 2D pose features. The DCSA module, while effective for feature fusion, primarily reuses self-attention and cross-attention mechanisms, limiting its conceptual novelty.
2. Insufficient motivation and justification: The motivation for introducing quantized features remains unclear. First, can the quantized features maintain consistency with the 2D poses from different viewpoints? Second, the input settings differ between the first-stage learning (single action input) and the second-stage learning (multi-view and multi-sequence input)—could this discrepancy amplify feature misalignment and introduce additional noise?
3. The experimental setup has several questionable aspects. First, the paper fails to provide an evaluation of the first-stage model's learning performance. Moreover, since the performance improvement is marginal, the authors should further validate—through visualization or other means—whether the proposed method genuinely addresses the challenges in camera parameter-free settings.

---

> ### Author Rebuttal · Authors · 2025-07-31
>
> > **Q1: Limited methodological novelty: This work proposes an incremental enhancement to existing methods by adding quantized 2D pose features. The DCSA module, while effective for feature fusion, primarily reuses self-attention and cross-attention mechanisms, limiting its conceptual novelty.**
>
> **A1:** We are the first to introduce a data-drive discrete pose prior into continous 3D pose regression for noise resilience, and to propose to fuse discrete and continuous features with complementary strengths. The use of attention for fusion is not very novel but intuitive and effective. However, **the novelty stems from the insights of combining data-drive robust discrete priors with precise continuous representations to improve noise robustness and regression precsion, not from a single component.** As shown in **Figure 3** of the Supplementary Material, our model outperforms the pure-continuous baseline by up to **5 mm** under increasing noise levels. Across all noise intensities, our method consistently achieves lower MPJPE than the baseline, demonstrating its strong noise resilience.
>
>
> > **Q2: Insufficient motivation and justification: The motivation for introducing quantized features remains unclear. First, can the quantized features maintain consistency with the 2D poses from different viewpoints? Second, the input settings differ between the first-stage learning (single action input) and the second-stage learning (multi-view and multi-sequence input)—could this discrepancy amplify feature misalignment and introduce additional noise?**
>
> **A2:**
> **(1) Viewpoint consistency is neither required nor expected for quantized features.**
> The quantized features are not designed to maintain consistency across viewpoints. In fact, enforcing such consistency is neither feasible nor necessary. For example, 2D poses from different views (e.g., front vs. side) can differ significantly in visible joints and geometric patterns, and forcing consistency may lead to loss of view-specific information during Stage I. Our goal is not to align cross-view 2D features, but to learn a **discrete pose prior** that captures **robust and generalizable patterns** within individual views, which is sufficient for supporting downstream regression.
>
> **(2) Stage I and Stage II use different inputs without causing feature misalignment.**
> Stage I uses single-view single-frame inputs to fully exploit large-scale unpaired 2D pose data, while Stage II leverages multi-view multi-sequence data for final 3D regression. This difference is intentional and serves complementary purposes: Stage I learns a **noise-robust discrete prior**, while Stage II captures **cross-view and temporal dependencies** through continuous features.
> There is **no misalignment**, as the discrete prior is not required to encode multi-view or temporal cues—it only needs to offer **robust structural guidance**. Multi-view and temporal modeling are deferred to the continuous branch, which is optimized jointly with the discrete tokens during Stage II, ensuring effective integration without conflict.
>
>
> > **Q3: The experimental setup has several questionable aspects. First, the paper fails to provide an evaluation of the first-stage model's learning performance.**
>
> **A3:** We report the performance of Stage I on AMASS in **Table 6 in Supplementary Material**. For single-frame, single-view evaluation on Human3.6M GT, Stage I achieves 40.2 MPJPE. It is important to note that the entire VQ-VAE contains only 5.54M parameters, which is quite small. And as expected, the discrete representation yields lower regression accuracy compared to continuous features. We will include those results in the main paper.
>
>
> > **Q4: Moreover, since the performance improvement is marginal, the authors should further validate—through visualization or other means—whether the proposed method genuinely addresses the challenges in camera parameter-free settings.**
>
> **A4:**
> **(1) Non-Marginal Performance Gains on Saturated Benchmarks.**
>
> (1.1) Saturation of Human3.6M and Stronger Results on 3DHP
>
> We would like to clarify that Human3.6M is already saturated in terms of performance, and Human3.6m contains some inaccurate annotations[4], making further improvement increasingly difficult.
> Unlike Human3.6M, which is a lab dataset, 3DHP includes both indoor and outdoor scenes, offering higher diversity and better reflecting real-world scenarios.
> Our method shows stronger performance on 3DHP without extra training data, with a **2.03 mm MPJPE reduction and a 4.24 AUC increase** over the second-best method, FusionFormer and SGraFormer.
>
> (1.2) Addressing the Overlooked Real-World Noise: A Novel Use of Data-Driven Discrete Priors for Robustness.
>
> Our core motivation is as follows: continuous representations are well-suited for regression tasks but are vulnerable to noise, while discrete representations tend to be noise-resilient but suffer in precision. Our method aims to leverage both to improve robustness without sacrificing accuracy. As shown in **Figure 3 of the supplementary material**, using DCSA and the discrete codebook improves noise robustness significantly, achieving up to **6% (5 mm MPJPE) improvement under large noise**, which is far from marginal.
>
> (1.3) Steady Performance Gains Through Progressive Discrete Prior Quality
>
> Table 4 (a)(c) shows **consistent and stage-wise improvements** when increasing the quality of the VQ-VAE prior (see L304–307, L322–324). These improvements are achieved without adding any model parameters, solely by enhancing the quality of the UniCodebook, which directly correlates the **quality of discrete prior with model robustness** and validating the effectiveness of our method, especially under noisy inputs.
>
> **(2) Enhancing Noise Robustness in Camera Parameter-Free Settings Which Is Important for Real-World Deployment**
>
> The concern you raised is exactly at the core of our work! In real-world camera parameter-free settings, noise is a big challenge. The core problem our method aims to address is noise sensitivity under uncalibrated conditions. Without camera extrinsics to provide geometric cues, such methods rely more heavily on semantic information from the data to perform multi-view fusion. This makes continuous-feature-based models, especially transformers, more vulnerable to noisy input data.
> To tackle this, we introduce discrete, data-driven human pose priors to support continuous regression. This improves robustness without sacrificing accuracy. As shown in **Figure 3 of the supplementary material**, our model consistently outperforms the continuous-only baseline under varying noise strengths and view counts, with up to a **5 mm (6.25%)** improvement under the highest noise level. This confirms that our method effectively mitigates the noise issue in camera-free settings.
>
>
> > **Q5: What are the differences in experimental settings between the first row of Table 4(a) and the baseline in Table 4(b) that lead to the performance discrepancy?**
>
> **A5:** Table 4 (a) T=1, Table 4 (b) T=27. Unless otherwise specified, all results in the paper default to T = 27. We apologize for not making this clear in the manuscript.
>
>
> > **Q6: The title format of Reference 32 is incorrect.**
>
> **A6:** We have corrected it.
>
>
> > **Q7: The authors should discuss how the inference efficiency compares with FusionFormer.**
>
> **A7:** We report the results in the table below, which shows that the computational cost of our model grows approximately linearly with the number of views and frames, consistent with Supplementary Table 4. Note that the reported parameter count is smaller than that in Figure 4 as we mistakenly included redundant components such as the 2D decoder, 3D encoder, and 3D decoder of VQVAE.
>
> | model        | Params (M) | MACs (G) v=1/2/3/4 and  f=1 | MACs(G) v=4 and f=3/9/27 |
> | ------------ | ---------- | --------------------------- | ------------------------ |
> | Fusionformer | 17.25      | 0.10/0.20/0.30/0.41         | 1.23/3.69/11.08          |
> | Ours         | 23.14      | 0.13/0.27/0.40/0.54         | 1.66/4.98/14.94          |

---

> > ### Author Response · Authors · 2025-08-04
> >
> > Dear Reviewer 63UD,
> >
> > We greatly appreciate the time and effort you have invested in reviewing our paper and providing insightful feedback. As a gentle reminder, it has been 4 days since we submitted our rebuttal. As the discussion period is drawing to a close, we wish to ensure that our rebuttal has comprehensively addressed your concerns. We are keen to receive any further feedback you might have and are prepared to make additional clarifications or modifications as needed. Thank you once again for your valuable insights. We look forward to your final thoughts.

---

> > > ### Author Response · Authors · 2025-08-08
> > >
> > > Dear Reviewer 63UD,
> > >
> > > We greatly appreciate the time and effort you have invested in reviewing our paper and providing insightful feedback. As a gentle reminder, it has been 8 days since we submitted our rebuttal. As the discussion period is drawing to a close, we wish to ensure that our rebuttal has comprehensively addressed your concerns. We are keen to receive any further feedback you might have and are prepared to make additional clarifications or modifications as needed. Thank you once again for your valuable insights. We look forward to your final thoughts.

---

> ### Comment · Area_Chair_QgdR · 2025-08-08
> **Final Call for Comments on Authors’ Rebuttal**
>
> Dear Reviewer 63UD,
>
> The Reviewer–Author Discussion period will end soon.
>
> Please review the authors’ rebuttal and share any comments or feedback so that the authors can provide additional responses if necessary.
>
> AC

---

### Official Review · Reviewer_Tfp8 · 2025-06-29

**Clarity:** 4
**Significance:** 4
**Originality:** 3
**Rating:** 5
**Confidence:** 4

**Summary:**

- This paper proposes a novel method for 3D human keypoint estimation from (potntially noisy) multi-view 2D keypoint sequences.
- They introduce *Uni-Codebook*, a noise-resilient discrete codebook that is trained to recover both 2D and 3D keypoints.
- They then use trained Uni-Codebook as a discrete prior for keypoint estimation, complementing and enhancing conventional estimation with spatio-temporal transformer. They term this Discrete-Continuous Spatial Attention (DCSA).
- They did experiments to demonstrate that by leveraging Uni-Codebook (trained on the AMASS dataset) as a prior, the proposed method achieves state-of-the-art results on the Human3.6M dataset.

**Questions:**

- You claim that your model is not affected by "predefined geometric relationships" (L283), yet you include view-dependent positional encoding $\mathbf{P}_{\mathrm {pos}}^V$ (L199). This appears to entangle positional relationship into the estimation process.
	- I would like to see if your method is truly independent from camera geometric relationships, for example, by swapping or reducing input views only during evaluation.
- You simply add the output of spatial MHSA and DCSA (Fig. 2, Eq. 9). Why wasn't a weighted sum used, as done in L236 for two-path integration?
	- The model benefits from adaptively considering continuous and discrete features depeneding on noise levels. Employing a weighted sum to allow the model to prioritize specific feature could lead to improved estimation.
	- Currently, since $\mathrm{head}_i$ are the sum of $V_i$ with the weights derived from $\mathrm{softmax} (QK)$ (which sums to one) for both of spatial MHSA and DCSA, this may hinder effective feature focusing.

**Ethical Concerns:**

["NO or VERY MINOR ethics concerns only"]

**Final Justification:**

My questions regarding some uncertainties about their methodology are fully answered by the authors. A minor issue about the necessity of positional encodings in their method is not yet fully experimented in the temporal model (T=27) setting due to the limited timeframe of the rebuttal period. However, as (1) they did an experiment in T=1, and the trend of the results should be the same for the temporal model, and (2) they assured they will include the result and discussion for the temporal experiment in the final paper, I am happy to keep my rating and accept this paper to NeurIPS 2025.

**Limitations:**

yes, in the supplementary materials

**Quality:**

4

**Strengths And Weaknesses:**

Strength:
- Uni-codebook offers a clear and concise solution for modality isolation.
	- It effectively combines the tokenization approach for pose estimation, previously used in TokenHMR [5] and PCT [9], and extends it to integrate 2D and 3D keypoint representations.
- They also propose DCSA, an intuitive and effective method to enhance continuous pose estimation with discrete representation.
- They achieve state-of-the-art results on the Human3.6M dataset. Comparisons and ablation studies are extensive and effectively supports their claims.

Weaknesses: integrated with the questions section.

---

> ### Author Rebuttal · Authors · 2025-07-31
>
> > **Q1: You claim that your model is not affected by "predefined geometric relationships" (L283), yet you include view-dependent positional encoding  (L199). This appears to entangle positional relationship into the estimation process. I would like to see if your method is truly independent from camera geometric relationships, for example, by swapping or reducing input views only during evaluation.**
>
> **A1:**
> (1) We acknowledge that our model does exhibit some sensitivity to view biases present in the data. However, this issue is not unique to our method and can also be observed in other SOTA approaches such as FusionFormer. Notably, our method demonstrates stronger generalization—even when trained using only 4-view data or a mixture of 3- and 4-view configurations—outperforming FusionFormer under the same conditions.
>
> We attribute this robustness to the use of discrete codebook features. Continuous representations are typically learned from multi-view data, where the same features must simultaneously encode both pose semantics and inter-view interactions. This entanglement can make them more sensitive to changing of view numbers. In contrast, our discrete features are learned from single-view data, and their integration into the model provides additional information that is less entangled with camera-specific biases.
>
>
> | model        | train views | test views | MPJPE    |
> | ------------ | :---------: | :--------: | -------- |
> | fusionformer |      4      |     4      | 27.0     |
> | fusionformer |      4      |     3      | 66.2     |
> | ours         |      4      |     4      | **26.3** |
> | ours         |      4      |     3      | **50.3** |
> | fusionformer |    4, 3     |     4      | 29.2     |
> | fusionformer |    4, 3     |     3      | 32.0     |
> | ours         |    4, 3     |     4      | **27.8** |
> | ours         |    4, 3     |     3      | **30.2** |
>
>
> (2) As attention mechanism is theoretically permutation-invariant, we suspect that this view sensitivity primarily stems from the use of positional encodings (PE). In ablation experiments where we removed PE from the model, we observed consistent performance improvements when shuffled views only when testing, suggesting that PE may inadvertently introduce view-dependent artifacts. But it still can not generalize to less views which is not observed in training. We believe that using reduce views when training would increate the performance.
>
> | model           | trian shuffle | train views | test shuffle | test views |  MPJPE   |
> | --------------- | :-----------: | :---------: | :----------: | :--------: | :------: |
> | ours            |       ×       |      4      |      ×       |     4      |   26.3   |
> | ours            |       ×       |      4      |      √       |     4      |   50.3   |
> | ours without PE |       ×       |      4      |      ×       |     4      |   26.3   |
> | ours without PE |       ×       |      4      |      √       |     4      | **26.3** |
> | ours without PE |       ×       |      4      |      ×       |     3      |   50.2   |
>
>
> > **Q2: You simply add the output of spatial MHSA and DCSA (Fig. 2, Eq. 9). Why wasn't a weighted sum used, as done in L236 for two-path integration? The model benefits from adaptively considering continuous and discrete features depending on noise levels. Employing a weighted sum to allow the model to prioritize specific feature could lead to improved estimation.**
>
> **A2:** Good suggestions! We apply weighted sum but obtained results similar to the original method when T=1. One possible explanation is that the model implicitly learns to balance the contributions from the two branches. Therefore, a straightforward addition already captures most of the benefits that a more complex weighted sum could provide, without adding extra parameters or complexity.
>
> | model               | MPJPE |
> | ------------------- | ----- |
> | ours                | 26.34 |
> | ours + weighted sum | 26.39 |
>
>
> > **Q3: Currently, since $head_i$ are the sum of $V_i$ with the weights derived from $softmax(QK)$ (which sums to one) for both of spatial MHSA and DCSA, this may hinder effective feature focusing.**
>
> **A3:**
> Excellent observation! In the original design, continuous queries interacted only with continuous keys/values, while a separate set of continuous queries interacted with discrete keys/values.
> However, this setup enforces that **a fixed number of continuous and discrete tokens are always involved in the output**, regardless of their actual feature quality or relevance in the current context. This can be limiting, since the relative importance of continuous and discrete features may vary depending on the context.
> To address this, we redesign the attention mechanism by unifying the two attention branches into a single one: continuous queries now attend to the concatenation of both continuous and discrete keys/values. This not only simplifies the architecture but also allows the model to dynamically and adaptively focus on the most relevant combination of continuous and discrete tokens, without being constrained by fixed ratios.
>
> However, preliminary experiments show that this unified attention approach does not improve performance—in fact, it slightly degrades it (MPJPE increases from 26.34 to 26.51).
> This might because that the two-path design inherently reduces coupling between branches, allowing each to specialize: one branch focuses on modeling continuous spatial pose relationships, while the other learns affinity to discrete tokens. This separation facilitates learning complementary features independently. In contrast, the single-path design requires the model to simultaneously capture both continuous spatial relationships and discrete token affinities within the same attention mechanism. This increased learning burden can cause feature interference and make it harder for the model to effectively disentangle and utilize the distinct modalities, leading to degraded performance.
>
> | model           | MPJPE |
> | --------------- | ----- |
> | ours (two path) | 26.34 |
> | ours (one path) | 26.51 |

---

> > ### Comment · Reviewer_Tfp8 · 2025-08-05
> > **Response to rebuttals**
> >
> > Thank you for your detailed rebuttal and I apologize for my late response.
> >
> > I agree with your rebuttal and have only one question. In your response A1(2), the table shows that your model performs MPJPE=26.3, but, according to Table 1 of the paper, this is 26.0, right? If it is indeed 26.3, it is the same as "ours without PE" and means your model performs the same with or without positional encodings.
> >
> > Other than that point, since all of my questions are fully answered, I'm happy to keep my rating and accept this paper.

---

> > > ### Author Response · Authors · 2025-08-05
> > >
> > > Thank you very much for your thoughtful follow-up and your willingness to accept the paper! We sincerely appreciate the time and effort you have dedicated to reviewing our work.
> > >
> > > Regarding your question, we apologize for the confusion. The MPJPE = 26.0 reported in Table 1 of the paper corresponds to the setting with **4 views and 27 frames**, whereas the MPJPE = 26.3 mentioned in the rebuttal refers to the setting with **4 views and only 1 frame**. This difference is clearly reflected in Figure 4(b) of the supplementary material, which compares model performance under varying frame counts.
> > >
> > > Since the number of frames does not affect the relative comparison between different model variants, and using fewer frames allows for faster experimentation, we adopted the frames=1 setting during the rebuttal period to conduct additional ablation studies more efficiently.
> > >
> > > Therefore, your observation is correct—**“Ours without PE” indeed becomes robust to shuffled view orders. Even though the model is trained with views in a fixed order, its performance remains stable when the view order is randomly shuffled during testing.** This highlights the effectiveness of removing positional encoding in promoting view-order invariance.
> > >
> > > Once again, thank you for your positive feedback and support—it truly means a lot to us!

---

> > > > ### Comment · Reviewer_Tfp8 · 2025-08-05
> > > > **Response to Authors**
> > > >
> > > > Thank you for the clarification. I apologize for my ignorance and for assuming the table value as a typo.
> > > >
> > > > I agree with your argument that the number of temporal frames shouldn't affect the relative performance. Training it with T=1 for rebuttal response for faster training makes sense.
> > > > The conclusion is nevertheless interesting --- It means your model without PE performs *better than with PE* with *slightly smaller number of parameters* (although the margin might be small) and becomes robust to shuffled views even though trained with fixed-order. Could this be because UniCodebook captures 2D-3D projection representation and doesn't require pose embedding?
> > > >
> > > > Anyway, if this trend is the same for multi-frame estimation (T=27), do you think the pose embedding is not necessary in the first place? Is it possible, by removing the pose embedding, to simplify the model and to strengthen your argument that your method is "not limited by predefined geometric relationships between views"?
> > > >
> > > > I do not ask additional experiments as discussion window is nearing the end, but happy to see the comparisons in the final paper.

---

> > > > > ### Author Response · Authors · 2025-08-06
> > > > >
> > > > > Thank you again for your thoughtful comments and encouraging feedback!
> > > > >
> > > > > Yes, we largely agree that the UniCodebook, by enforcing a shared space for both 2D and 3D data, may help the model capture implicit 2D–3D projection cues without relying on PE. This is a very interesting observation, and we will include additional experiments in the final version after acceptance to further verify and explore its potential.
> > > > >
> > > > > Regarding the view-order robustness, we sincerely thank you for bringing this to our attention. After removing PE, the model becomes highly robust to shuffled views, which appears to support our claim that it does not rely on predefined geometric relationships between views. Based on this, we plan to simplify the model by removing PE, and will benchmark it under shuffled-view and reduced-view settings against other calibration-free multi-view methods. We will include the analysis in the supplementary material of the final version.
> > > > >
> > > > > Thank you once again for your valuable support and suggestions!

---

### Official Review · Reviewer_iezF · 2025-07-02

**Clarity:** 3
**Significance:** 2
**Originality:** 3
**Rating:** 3
**Confidence:** 4

**Summary:**

This paper introduces a calibration-free multiview 3D human pose estimation method. It combines discrete and continuous representations using a unified VQ-VAE codebook (UniCodebook) and Discrete-Continuous Spatial Attention (DCSA). It uses discrete pose priors to improve the method's robustness against noisy 2D inputs while keeping the accuracy of continuous regression. Experimental results are shown on the Human3.6M dataset, as well as the MPI-INF-3DHP and FreeMan datasets.

**Questions:**

Is the codebook generalizable to unseen poses?

**Ethical Concerns:**

["Major Concern: Human rights (including surveillance)"]

**Final Justification:**

The rebuttal addressed some of my concerns, while still cannot conving me in the following points:
Comparison With Monocular Baselines Using Mixste Backbone: CPN 2D pose is already noise detection result comparing to GT 2D pose, I don't think it is necessary to add extra noise to CPN 2D. While with CPN 2D, the proposed method cannot outperform any baseline with the proposed robust prior.

Additional Monocular Evaluations With 27-Frame Input: The authors mention the proposed method "achieves the best monocular performance among all models originally designed for multi-view pose estimation, highlighting its strong generalization and potential for use even in monocular scenarios", while from its provided results, it cannot outperformance any monocular method under monocular setting. I still believe that the target multi-view setting limits the application of pose estimation methods, and how to apply them to real-world scenarios is still worth more in-depth discussion.

In the discussion period, the author try to convince reviewers by some non-standard settings such as "train-on-GT, test-on-CPN setting", I encourge the authors use the standard comparison in the futher submission version.

**Limitations:**

Assumptions of a multiview setting limit the application of this method. How to apply it on real scenarios, such as moving cameras or video streams?

**Paper Formatting Concerns:**

No qualitative evaluation and comparison with baselines are shown in the main paper, which makes it very difficult to show the improvement convincingly.

**Quality:**

2

**Strengths And Weaknesses:**

Strengths:

S1: It is novel to unify 2D/3D pose quantization in a shared codebook, mitigating the modality gap. DCSA elegantly resolves the discrete-continuous mismatch via soft attention. The rigorous two-stage training (codebook pretraining + transformer lifting) is reasonable.

S2: The ablations validate design choices (Table 4) are very helpful.

S3: The result shows the proposed method outperforms both recent calibrated (e.g., CrossView Fusion) and calibration-free (e.g., FusionFormer) methods.

Weaknesses:

W1: Missing baselines, especially for the monocular setting, only baselines are listed; it should also evaluate this method's monocular performance on different datasets.

W2: The fixed-size UniCodebook may have difficulty with highly diverse or unusual poses that were not included in the training. This creates a representational bottleneck that might limit generalization.

W3: The paper does not explain how the codebook would adjust to new pose domains. Needing complete retraining could be impractical in real-world situations.

---

> ### Author Rebuttal · Authors · 2025-07-31
>
> > **Q1: Missing baselines, especially for the monocular setting, only baselines are listed; it should also evaluate this method's monocular performance on different datasets.**
>
> **A1:**
> (1) Regarding monocular performance, we are currently training our temporal model on sequences of 243 frames, which requires approximately 3 days to converge. Once training is complete, we will report monocular results and analyze the effect of the discrete codebook on noise robustness in the temporal monocular setting within the rebuttal period.
>
> (2) Our method is primarily designed for the multi-view and temporal setting (v=4, f=27), which is a meaningful and practical setup. Multi-view configurations are as important as monocular ones, as they better reflect real-world applications such as motion capture studios and multi-camera systems, where complementary viewpoints enhance robustness and accuracy.
>
> (3) Notably, recent state-of-the-art multi-view, uncalibrated methods like SGraFormer and SVTFormer also do not report monocular performance, as their architectures and training are specialized for multi-view input.
>
>
> > **Q2:The fixed-size UniCodebook may have difficulty with highly diverse or unusual poses that were not included in the training. This creates a representational bottleneck that might limit generalization. Is the codebook generalizable to unseen poses?**
>
> **A2:**
> (1) We agree that a fixed-size UniCodebook may not fully cover all highly diverse or unusual poses. **However, continuous regression models face the same limitation.**
>
> (2) **Noise in real-world scenarios can also hinder generalization, so improving noise robustness directly contributes to better generalization.** As shown in **Figure 3 of the supplementary material**, when noise intensity increases, the poses become more unseen for the UniCodebook (as traing pose is high quality from AMASS). In this case, our model are more robust than the continuous baseline.
>
> (3) **Better data utilization also improves real-world generalization.** The UniCodebook is trained on high-quality poses from AMASS and supports both paired and unpaired inputs, allowing broader data utilization than Stage II .
> This means that Stage I, based on the UniCodebook, is less likely to become a bottleneck—especially considering that the continuous model in Stage II is both more sensitive to noise and trained on less data.
>
> (4) To address generalization concerns, we propose DCSA to effectively combine the complementary strengths of discrete and continuous representations.
> This hybrid design ensures that our model does not suffer from more domain generalization issues than fully continuous alternatives.
>
>
> > **Q3: The paper does not explain how the codebook would adjust to new pose domains. Needing complete retraining could be impractical in real-world situations.**
>
> **A3:**
> (1) As shown in Section F of the supplementary material, our model encounters challenges primarily when transferring across different **pose definitions**, rather than across different **pose distribution domains**.
>
> In our work, a pose is defined by a set of semantically consistent keypoints representing major human joints, their 3D coordinates, and the skeletal connectivity between them. For example, in the Human3.6M dataset, each pose comprises 17 keypoints—including the head, neck, shoulders, elbows, wrists, hips, knees, and ankles—structured as a kinematic chain.
>
> It is important to distinguish this from domain shifts in pose distributions, which are naturally involved in our main experiments—for example, transferring from AMASS to Human3.6M and other datasets.
> **We highlight this distinction to clarify a potential misunderstanding: the reviewer’s concern appears to conflate variation in pose definitions (i.e., joint types and connections) with shifts in pose distribution (i.e., statistical variation within a fixed definition).**
>
> (2) As shown in **Section F**, we propose a potential solution by training a format-agnostic discrete prior. However, we currently do not have a concrete approach for fine-grained adaptation to new pose definitions, as this is not the primary focus of our work. **This limitation is common to all models that represent poses as 1D sequences, and it is not specific to the use of VQ-VAE.**
>
> (3) Moreover, **many joint formats can be easily converted**, especially when migrating from models with more joints to those with fewer joints. Thus, retraining is often unnecessary unless the joint sets are truly incompatible. In real applications, **models typically rely on mainstream 2D detectors, which use fixed and widely adopted pose definitions.** For example, OpenPose, Detectron2 and MMPose all support COCO format keypoint.
>
>
>
>
> > **Q4: No qualitative evaluation and comparison with baselines are shown in the main paper, which makes it very difficult to show the improvement convincingly.**
>
> **A4:** We include qualitative evaluations and comparisons with baselines in **Figure 7 and the accompanying videos in the Supplementary Materials**. In the final version, we will move part of these qualitative results into the main paper and provide more detailed analysis to better highlight the improvements achieved by our method.
>
>
> > **Q5: Assumptions of a multiview setting limit the application of this method. How to apply it on real scenarios, such as moving cameras or video streams?**
>
> **A5:**
> (1) As noted in our response to Q1, our model can be directly applied to monocular settings by setting views=1, while still maintaining robustness to noisy 2D pose inputs (see **Figure 3 in Supplementary Materials**).
> Although our main experiments use a multiview setup, we share the same concern as the reviewer regarding real-world deployment, where noisy 2D poses are common, especially in moving camera scenarios.
>
> (2) Uncalibrated multiview scenarios are susceptible to noisy 2D poses and thus present a realistic and challenging setting. Without camera extrinsics to provide geometric cues, such methods rely more heavily on semantic information for multi-view fusion. This makes continuous-feature-based models more vulnerable to input noise. Our study on robustness under uncalibrated multiview conditions is therefore also helpful to handle temporal jitter caused by moving cameras, offering practical insights for deployment in dynamic real-world environments.

---

> > ### Author Response · Authors · 2025-08-02
> > **Supplmetanry Response to Q1**
> >
> > > **Q1: Missing baselines, especially for the monocular setting, only baselines are listed; it should also evaluate this method's monocular performance on different datasets.**
> >
> > **A1:**
> >
> > **(1) Comparison With Monocular Baselines Using Mixste Backbone:**
> > Our continuous backbone has significantly fewer parameters than typical monocular baselines (e.g., Ours: 17M vs. MixSTE: 33M), and our method lost tricks for temporal modeling. To better assess temporal performance under a fair setting, we adopt the stronger MixSTE backbone for our method and conduct experiments with 243-frame sequences.
> >
> > The results show that although our performance is slightly lower than MixSTE, **our method demonstrates significantly stronger robustness under noisy conditions (achieving a 15.8 MPJPE, a 16.8% relative improvement)**. This supports a key finding of our paper: continuous representations are vulnerable to noise, and incorporating discrete priors can markedly improve robustness.
> >
> > | model             | frames | input          |  MPJPE   |
> > | ----------------- | :----: | -------------- | :------: |
> > | MixSTE [34]       |  243   | CPN 2D         | **40.9** |
> > | MixSTE [34]       |  243   | CPN 2D + noise |   93.9   |
> > | MixSTE + codebook |  243   | CPN 2D         |   42.7   |
> > | MixSTE + codebook |  243   | CPN 2D + noise | **78.1** |
> >
> > **(2) Additional Monocular Evaluations With 27-Frame Input:**
> > Due to the high computational cost of training with 243-frame sequences, we further validate our method using shorter 27-frame sequences. Results marked with † denote our re-implementation, while others are taken directly from the original papers.
> >
> > As shown in table below, **among all methods originally designed for multi-view 3D human pose estimation, our method achieves the best performance under monocular settings**, despite not being explicitly designed for this setup.
> >
> > While our model does not surpass all state-of-the-art methods tailored for monocular settings, it still achieves competitive performance (achieving 46.2 mm MPJPE vs. 44.1 mm MPJPE of STCFormer).
> >
> > | model            | design for multiview? | frames | MPJPE |
> > | ---------------- | :-------------------: | :----: | :---: |
> > | MHFormer [17]    |                       |   27   | 45.9  |
> > | MixSTE [34]      |                       |   27   | 45.1  |
> > | STCFormer [40]   |                       |   27   | **44.1**  |
> > | MTF [26]         |           √           |   27   | 49.1  |
> > | fusionformer [1] † |           √           |   27   | 51.3  |
> > | ours             |           √           |   27   | **46.2**  |
> >
> >
> > **(3) Summary :**
> >
> > (3.1) As shown in Table 1, our method achieves state-of-the-art results under the multi-view setting (view=4, frame=27). **We argue that improvements in multi-view settings are equally valuable as those in monocular settings**, especially given the growing deployment of multi-camera systems.
> >
> > (3.2) Although we do not achieve the best temporal performance, **our incorporation of discrete priors and the proposed DCSA module significantly enhance robustness to noise**—a challenge that has been largely overlooked in previous work. This constitutes a core contribution of our paper.
> >
> > (3.3) Finally, **our method achieves the best monocular performance among all models originally designed for multi-view pose estimation**, highlighting its strong generalization and potential for use even in monocular scenarios.
> >
> > [40] 3d human pose estimation with spatio-temporal criss-cross attention. CVPR 2023

---

> > > ### Author Response · Authors · 2025-08-04
> > >
> > > Dear Reviewer iezF,
> > >
> > > We greatly appreciate the time and effort you have invested in reviewing our paper and providing insightful feedback. As a gentle reminder, it has been 4 days since we submitted our rebuttal. As the discussion period is drawing to a close, we wish to ensure that our rebuttal has comprehensively addressed your concerns. We are keen to receive any further feedback you might have and are prepared to make additional clarifications or modifications as needed. Thank you once again for your valuable insights. We look forward to your final thoughts.

---

> > > > ### Comment · Reviewer_iezF · 2025-08-05
> > > >
> > > > (1) Comparison With Monocular Baselines Using Mixste Backbone:
> > > > CPN 2D pose is already noise detection result comparing to GT 2D pose, I don't think it is necessary to add extra noise to CPN 2D. While with CPN 2D, the proposed method cannot outperform any baseline with the proposed robust prior.
> > > >
> > > > (2) Additional Monocular Evaluations With 27-Frame Input:
> > > > The authors mention the proposed method "achieves the best monocular performance among all models originally designed for multi-view pose estimation, highlighting its strong generalization and potential for use even in monocular scenarios", while from its provided results, it cannot outperformance any monocular method under monocular setting. I still believe that the target multi-view setting limits the application of pose estimation methods, and how to apply them to real-world scenarios is still worth more in-depth discussion.

---

> > > > > ### Author Response · Authors · 2025-08-08
> > > > >
> > > > > Thank you for your patience. We apologize for the delay as some of the additional experiments took time to complete.
> > > > >
> > > > > We fully acknowledge your insightful comment regarding the limited improvement of our method under the temporal setting. Indeed, our codebook-based representation does not provide significant gains in sequence modeling, and we truly appreciate this observation. However, we respectfully ask the reviewer to consider two key aspects of our model:
> > > > >
> > > > > > **1. Consistent Improvement in Multi-View Settings**:
> > > > >
> > > > > As shown in the table below, our model demonstrates clear and stable improvements when the number of views increases from 2 to 4. This indicates our method's stable effectiveness in multi-view fusion, with potential reasons explained in Point 3.
> > > > >
> > > > > |model|dataset|view|MPJPE|
> > > > > |:-|:-:|:-:|-:|
> > > > > |baseline|h36m|2|43.11|
> > > > > |baseline + codebook|h36m|2|**42.13**|
> > > > > |baseline|h36m|3|29.90|
> > > > > |baseline + codebook|h36m|3|**29.46**|
> > > > > |baseline|h36m|4|27.01|
> > > > > |baseline + codebook|h36m|4|**26.34**|
> > > > >
> > > > > > **2. Superior Generalization in Monocular Real-World Settings**:
> > > > >
> > > > > While our model may underperform slightly on clean GT 2D inputs compared to state-of-the-art transformer baselines, it shows **clear advantages under realistic deployment conditions**.
> > > > >
> > > > > In real-world applications, ground-truth 2D poses are unavailable, and detector predictions are affected by noise from occlusion, blur, and lighting—challenges not reflected in clean benchmark data.
> > > > > **Rather than adding artificial noise, we simulate a practical scenario where the model is trained on GT 2D poses but tested with 2d detector predictions (e.g., from CPN)**.
> > > > >
> > > > > Results under different train/test combinations (GT->GT, CPN->CPN, etc, where the notation indicates training on the former and testing on the latter) primarily reflect a model’s capacity to fit data.
> > > > >
> > > > > In real applications, models are often trained with abundant, high-quality 2D–3D data, yet deployed with static, frame-by-frame 2D pose estimators that lack temporal context and are sensitive to occlusion and motion blur.
> > > > > Therefore, the **train-on-GT, test-on-CPN** setting serves as a more reasonable protocol for real-world performance, where the model must generalize from ideal supervision to noisy downstream conditions.
> > > > >
> > > > > |model|train 2d|frame|test MPJPE (gt 2d)|test MPJPE (cpn 2d) (close to real-world setting)|
> > > > > |:-|:-:|:-:|:-:|:-:|
> > > > > |MixSTE|GT|243|21.6|64.6|
> > > > > |KTPFormer|GT|243|19.0|66.6|
> > > > > |ours|GT|243|23.1|**52.8(20.7%↓)**|
> > > > >
> > > > > As shown, **our model achieves a 13.8mm (20.7%) improvement over the best-performing baseline (KTPFormer)**, which shows its stable and reliable result in practical applications. Unlike minor benchmark differences (e.g., 2–3mm), this margin represents a **clearly visible improvement in practical use**.
> > > > >
> > > > > Moreover, we observe that **transformer-based models that perform better on clean inputs often degrade more severely when tested with noisy 2D poses**. For example, KTPFormer achieves the lowest MPJPE on GT input but suffers the worst drop in performance when tested with CPN detections. This highlights their limited robustness and suggests potential overfitting to clean training distributions. In contrast, our model demonstrates much stronger **noise resilience and generalization**, making it especially valuable for **real-world settings such as moving-camera scenarios, in-the-wild videos, or mobile applications**, where temporal jitter and noise are prevalent.
> > > > >
> > > > > In short, while SOTA models may excel under ideal conditions, our method delivers **more consistent and reliable performance where it truly matters—under noisy, real-world conditions**.

---

> > > > > > ### Author Response · Authors · 2025-08-08
> > > > > >
> > > > > > > **3. Why Does the Codebook Help in Multi-View but Not in Temporal Settings?**
> > > > > >
> > > > > > We believe the different behavior of the codebook across these two settings arises from the distinct characteristics of the data:
> > > > > >
> > > > > > **(1) Multi-View Fusion Benefits From Unified Representation**:
> > > > > >
> > > > > > In the multi-view setting, the model needs to fuse **highly diverse 2D pose observations** from different viewpoints, which can be extremely heterogeneous in appearance due to camera perspective distortion. For instance, in the Human3.6M dataset, the average MPJPE between 2D poses captured from two different views of the same 3D frame is **161 pixels MPPJE**, reflecting substantial cross-view discrepancies in image-space keypoints.
> > > > > >
> > > > > > Such discrepancies pose a significant challenge for multi-view fusion. Although these 2D poses all correspond to the same 3D pose, their visual difference makes direct feature-level fusion difficult without additional constraints or guidance.
> > > > > >
> > > > > > This is where our **UniCodebook-based representation** proves especially effective. During training, the UniCodebook learned using paired 2D–3D information across **multiple views**, allowing it to encode **a unified latent space** where different 2D projections of the same 3D pose are mapped to similar discrete representations. As a result:
> > > > > > - Features extracted from different views are naturally **aligned in the latent space**, even if their 2D inputs differ significantly.
> > > > > > - This alignment makes it easier for the model to **fuse complementary information** from different perspectives, as the network no longer needs to learn complex view-specific transformations from scratch.
> > > > > > - It also reduces the risk of feature-level conflict or inconsistency when aggregating information across views.
> > > > > >
> > > > > > **(2) Temporal Fusion Relies on Subtle Motion Cues**:
> > > > > >
> > > > > > In contrast, sequential 2D poses from the same camera are temporally smooth, with much smaller inter-frame differences, e.g., an average MPJPE of only **1.03 pixels MPJPE** between adjacent frames. These subtle motion cues are well captured by continuous representations. However, when discretized into codebook tokens, these fine-grained variations may lost, thereby limiting the usefulness of the codebook in modeling motion continuity.
> > > > > >
> > > > > > > **4. Summary**
> > > > > >
> > > > > > We are very grateful to the reviewer for raising this important issue.
> > > > > > We will include these additional results and analyses in the final version of the paper, highlighting both the strengths and limitations of our method.
> > > > > > If you have other questions or concerns, please don't hesitate to discuss with us.

---

### Official Review · Reviewer_imRL · 2025-07-04

**Clarity:** 2
**Significance:** 2
**Originality:** 2
**Rating:** 4
**Confidence:** 4

**Summary:**

This paper propose a model for uncalibrated multi-view human pose estimation. The model has two part, a VQ-VAE and a regressional transformer that leverages the 2D poses and pre-trained priors in VQ-VAE to compute 3D human poses, for which the authors claimed for some architectural novelties. The evaluation is done on several 3D human pose estimation benchmarks, on which the proposed method outperforms some previous works.

**Questions:**

N/A

**Ethical Concerns:**

["NO or VERY MINOR ethics concerns only"]

**Final Justification:**

I have read the author rebuttal and other reviewer's comments. I think the proposed method's good performance under heavy occlusion is beneficial to the community, for which I ignored in my first read, thus I am happy to raise my rating.

**Quality:**

2

**Strengths And Weaknesses:**

**Strengths**
* The idea of leveraging pre-trained priors in VQ-VAE is technically sound
* The experiment setting is good and comprehensive enough. Ablation studies are presented to test the influence of each proposed modules.

**Weakness**
* My biggest concern for this work is that it seems all its proposed modules only have marginal influence on the model performance. From the results in Table 4 (a), (b), (c), I do not think there any significant improvement resulted from the "new ideas", e.g., the MPJPE improvements are all less then 3% compared to the baseline. For me, they could just be random noises.
* When comparing with state-of-the-art, e.g., Table 1, the author should compare against latest methods instead of out-of-date ones. For example, for Human3.6M, I recommend the authors to compare methods listed in this list: https://paperswithcode.com/sota/3d-human-pose-estimation-on-human36m.
* Some technical details are not clear. For example, in L177, how are the $N$ embeddings computed from the $J$ joints? Why choose $N$ to be 63? It looks like a magic number to me.
* For the VQ-VAE, why not adding temporal modelling? My gut tells me maybe adding temporal modelling will give some performance boost.
* For L209-L230, the authors spend a lot of spaces to introduce the basic attention formulation, for which I do not think it's necessary. TBH, although it does make the paper look more "mathematical" at the first glance, it does not bring any real values to the readers given the attention and transformers are widely used today. Please do not expect reviewers are fool enough to give accept just because there are some "nice looking" formulations for well-known basic stuffs :))

---

> ### Author Rebuttal · Authors · 2025-07-31
>
> > **Q1: My biggest concern for this work is that it seems all its proposed modules only have marginal influence on the model performance. From the results in Table 4 (a), (b), (c), I do not think there any significant improvement resulted from the "new ideas", e.g., the MPJPE improvements are all less then 3% compared to the baseline. For me, they could just be random noises.**
>
> **A1:** **(1) Saturation of Human3.6M and Stronger Results on 3DHP**
>
> We would like to clarify that Human3.6M is already saturated in terms of performance, and Human3.6m contains some inaccurate annotations[4], making further improvement increasingly difficult.
> Unlike Human3.6M, which is a lab dataset, 3DHP includes both indoor and outdoor scenes, offering higher diversity and better reflecting real-world scenarios.
> Our method shows stronger performance on 3DHP without extra training data, with a **2.03 mm MPJPE reduction and a 4.24 AUC increase** over the second-best method, FusionFormer and SGraFormer.
>
> **(2) Addressing the Overlooked Real-World Noise: A Novel Use of Data-Driven Discrete Priors for Robustness**
>
> Our core motivation is as follows: continuous representations are well-suited for regression tasks but are vulnerable to noise, while discrete representations tend to be noise-resilient but suffer in precision. Our method aims to leverage both to improve robustness without sacrificing accuracy. As shown in **Figure 3 of the supplementary material**, using DCSA and the discrete codebook improves noise robustness significantly, achieving up to **6% (5 mm MPJPE) improvement under large noise**, which is far from marginal.
>
> **(3) Steady Performance Gains Through Progressive Discrete Prior Quality**
>
> Table 4 (a)(c) shows **consistent and stage-wise improvements** when increasing the quality of the VQ-VAE prior (see L304–307, L322–324). These improvements are achieved without adding any model parameters, solely by enhancing the quality of the UniCodebook, which directly correlates the **quality of discrete prior with model robustness** and validating the effectiveness of our method, especially under noisy inputs.
>
>
> > **Q2: When comparing with state-of-the-art, e.g., Table 1, the author should compare against latest methods instead of out-of-date ones. For example, for Human3.6M, I recommend the authors to compare methods listed in paperwithcode.**
>
> **A2:** (1) Thank you for the suggestion. **We have compared our method against all latest open-source available models under the uncalibrated setting** (e.g., SVTFormer (AAAI 2025), SGraFormer (AAAI 2024), FusionFormer(AAAI 2024)).
>
> (2) We followed your suggestion and compared our method with the top-3 approaches listed on Papers With Code. Methods ranked lower show performance significantly inferior to ours.
>
> (2.1) In that leaderboad, the top-ranked method LMT R152[1] operates under a different experimental setup by generating 3D poses directly from images and leveraging SMPL prior, which is **not comparable to our setting** that uses noisy 2D pose inputs. As noted in the main paper (Lines 247–249), the 2D poses predicted by CPN contain significant noise, making it challenging to infer correct 3D poses from erroneous 2D joints. In contrast, image-based methods inherently have access to more accurate 2D information.
>
> (2.2) The second-ranked method, Geometry-Biased Transformer[2], relies on calibrated setups and uses HRNet for 2D pose sequences. Since HRNet’s 2D pose estimation is significantly more accurate than CPN’s, it benefits from better inputs. Despite using noisier inputs and lacking camera calibration, our method achieves comparable results (MPJPE: 26.0 mm VS 26.0 mm), highlighting the robustness and strength of our approach. This also suggests the dataset performance is approaching saturation.
>
> (2.3) Moreover, our method substantially outperforms the third-ranked Epipolar Transformer[3] (26.0 mm vs. 26.9 mm). We will include these methods in the final version and clarify the differences in experiment settings between our method and theirs.
>
> | model                           | calibrate | input                                         | MPJPE |
> | ------------------------------- | :-------: | --------------------------------------------- | :---: |
> | LMT R152 [1]                    |     √     | image -> 3D pose                              |   \   |
> | Geometry-Biased Transformer [2] |     √     | HRNet 2D pose sequence -> 3D pose             | 26.0  |
> | Epipolar Transformer [3]        |     √     | image -> 3D pose                              | 26.9  |
> | ours                            |     ×     | CPN 2D pose sequence -> 3D pose               | 26.0  |
>
>
> > **Q3: Some technical details are not clear. For example, in L177, how are the $N$ embeddings computed from the $J$ joints? Why choose $N$ to be 63? It looks like a magic number to me.**
>
> **A3:** We apologize for not explaining the origin of this number. Human3.6M contains 17 joints, each with 3D coordinates, totaling 51 dimensions. To increase representation redundancy, we use 63 tokens, roughly corresponding to 21 joints × 3 coordinates. 63 is not a magic number as similar token counts (e.g., 60 or 65) yield similar performance.
>
> | token num | MPJPE |
> | :-------: | ----- |
> | 60        | 26.09 |
> | 63        | 26.04 |
> | 65        | 26.07 |
>
>
> > **Q4: For the VQ-VAE, why not adding temporal modelling? My gut tells me maybe adding temporal modelling will give some performance boost.**
>
> **A4:** (1) Thank you for the insightful suggestion. As have discussed in **Section F in Supplementary Material**, we fully agree that incorporating temporal modeling can potentially enhance performance. However, we intentionally avoid relying on pose sequences in order to maximize the use of single-pose data, which is significantly more abundant than sequential data.
>
> (2) We also conducted experiments incorporating a temporal prior during codebook training by introducing a loss term that encourages the predicted joint velocity across time to match that of the ground truth if two joints are from one sequence. This indeed led to improvements in VQ-VAE reconstruction across all four codebook strategies. When applied to the full backbone, the final MPJPE remained almost unchanged (26.09 mm).
>
> | model                 | 2Dto2D | 2Dto3D | 3Dto2D | 3Dto3D |
> | --------------------- | ------ | ------ | ------ | ------ |
> | ours                  | 0.006  | 28.81  | 0.005  | 7.84   |
> | ours + temporal VQVAE | 0.006  | 28.31  | 0.005  | 7.57   |
>
> | model                 | Stage II MPJPE |
> | --------------------- | -------------- |
> | ours                  | 26.02          |
> | ours + temporal VQVAE | 26.09          |
>
>
> > **Q5: For L209-L230, the authors spend a lot of spaces to introduce the basic attention formulation, for which I do not think it's necessary. TBH, although it does make the paper look more "mathematical" at the first glance, it does not bring any real values to the readers given the attention and transformers are widely used today. Please do not expect reviewers are fool enough to give accept just because there are some "nice looking" formulations for well-known basic stuffs**
>
> **A5:** Thanks! We will simplify those redundant explanation of the basic attention mechanism and focus directly on the core ideas and contributions of our method.
>
> [1] Learnable Human Mesh Triangulation for 3D Human Pose and Shape Estimation. WACV 2023
>
> [2] Geometry-Biased Transformer for Robust Multi-View 3D Human Pose Reconstruction. FG 2024
>
> [3] Epipolar Transformers. CVPR 2020
>
> [4] Learnable Triangulation of Human Pose. ICCV 2019

---

> > ### Author Response · Authors · 2025-08-04
> >
> > Dear Reviewer imRL,
> >
> > We greatly appreciate the time and effort you have invested in reviewing our paper and providing insightful feedback. As a gentle reminder, it has been 4 days since we submitted our rebuttal. As the discussion period is drawing to a close, we wish to ensure that our rebuttal has comprehensively addressed your concerns. We are keen to receive any further feedback you might have and are prepared to make additional clarifications or modifications as needed. Thank you once again for your valuable insights. We look forward to your final thoughts.

---

> > > ### Author Response · Authors · 2025-08-08
> > >
> > > Dear Reviewer imRL,
> > >
> > > We greatly appreciate the time and effort you have invested in reviewing our paper and providing insightful feedback. As a gentle reminder, it has been 8 days since we submitted our rebuttal. As the discussion period is drawing to a close, we wish to ensure that our rebuttal has comprehensively addressed your concerns. We are keen to receive any further feedback you might have and are prepared to make additional clarifications or modifications as needed. Thank you once again for your valuable insights. We look forward to your final thoughts.

---

> ### Comment · Area_Chair_QgdR · 2025-08-08
> **Final Call for Comments on Authors’ Rebuttal**
>
> Dear Reviewer imRL,
>
> The Reviewer–Author Discussion period will end soon.
>
> Please review the authors’ rebuttal and share any comments or feedback so that the authors can provide additional responses if necessary.
>
> AC

---

### Note · Authors · 2025-08-13

> **Summary of our work**

We propose leveraging robust data-driven discrete priors to mitigate noise in multi-view 3D human pose estimation. Our method integrates discrete priors with continuous representations using the DCSA mechanism, improving noise robustness without sacrificing regression accuracy, as consistently validated across diverse benchmarks and noise conditions.

> **Recognized strengths by reviewers**

1. Novel, technically sound, and well-motivated integration of noise-robust discrete priors into a continuous backbone to enhance noise robustness. The UniCodebook further improves data efficiency and reduces the 2D-3D gap. (imRL, iezF, Tfp8)

2. Comprehensive experiments validating each design choice. (imRL, iezF, Tfp8, 63UD)

3. Superior performance over recent multi-view methods across multiple datasets. (iezF, Tfp8)

> **Review feedback and our responses**

1. Marginal Improvement (imRL, 63UD): Human3.6M is saturated; our method gains on 3DHP (**+2.03 mm MPJPE, +4.24 AUC**) and up to **6% MPJPE reduction under heavy noise**, demonstrating robust improvements for real-world noisy inputs.

2. Limited Novelty (63UD): **Our novelty lies in combining data-driven robust discrete priors with precise continuous representations to improve noise robustness and regression accuracy, not from a single component.** This is validated by consistent benchmark improvements and acknowledged by other three reviewers.

3. Missing Baselines (imRL, iezF):

- For multiview methods, all public calibration-free baselines and relevant Papers with Code methods compared; **ours remains SOTA**.

- For monocular methods, we report only a 1–3 mm MPJPE gap to temporal SOTA. Applying other uncalibrated multi-view models to single-view input, **our method outperforms them**.

- Importantly, in training-on-clean/testing-on-noisy scenarios, we outperform monocular SOTA by **13.8 mm MPJPE (20.7%)**, confirming practical effectiveness.

Following these clarifications, reviewer Tfp8 retained a high score. Reviewer iezF participated in one round, to which we responded with detailed results and analyses. For reviewers 63UD and imRL, who did not engage in discussion, we provide detailed point-by-point responses. We fully understand that they may have competing commitments limiting follow-up participation.

We will address all concerns and incorporate reviewers’ suggestions to further improve our paper. We thank the AC and reviewers for their thoughtful evaluation and engagement.

---

### Decision · Program_Chairs · 2025-09-17

**Decision:**

Accept (poster)

**Comment:**

The paper presents a multi-view 3D human pose estimation method considering a noisy and calibration-free setup by leveraging data-driven discrete priors. The proposed method shows the SOTA performance.

The paper initially received very mixed ratings. However, after the rebuttal and author–reviewer discussion phase, the authors provided thorough responses with additional verifications. As a result, two reviewers raised their scores, leading to three acceptance ratings and one rejection rating.

The AC acknowledges that the proposed noise-robust discrete priors demonstrate clear efficacy in the target scenarios and that the paper provides sufficient evidence of its strengths. Given this, the AC recommends acceptance.

The rebuttal was particularly important in convincing reviewers, and the authors are strongly encouraged to revisit it and incorporate it into their revision. In addition, the final comments by Reviewer iezF are highly valuable, and the authors should carefully reflect on this feedback.